# Dynamic coupling of fast channel gating with slow ATP-turnover underpins protein transport through the Sec translocon

Joel A Crossley [1,2,3], William J Allen [4], Daniel W Watkins [4], Tara Sabir [3], Sheena E Radford [1], Roman Tuma [1,2], Ian Collinson [4✉] & Tomas Fessl [2✉]

## Abstract

**The Sec translocon is a highly conserved membrane assembly for polypeptide transport across, or into, lipid bilayers. In bacteria, secretion through the core channel complex—SecYEG in the inner membrane—is powered by the cytosolic ATPase SecA. Here, we use single-molecule fluorescence to interrogate the conformational state of SecYEG throughout the ATP hydrolysis cycle of SecA. We show that the SecYEG channel fluctuations between open and closed states are much faster (~20-fold during translocation) than ATP turnover, and that the nucleotide status of SecA modulates the rates of opening and closure. The SecY variant PrlA4, which exhibits faster transport but unaffected ATPase rates, increases the dwell time in the open state, facilitating pre-protein diffusion through the pore and thereby enhancing translocation efficiency. Thus, rapid SecYEG channel dynamics are allosterically coupled to SecA via modulation of the energy landscape, and play an integral part in protein transport. Loose coupling of ATP-turnover by SecA to the dynamic properties of SecYEG is compatible with a Brownian-rachet mechanism of translocation, rather than strict nucleotide-dependent interconversion between different static states of a power stroke.**

**Keywords** Dynamic Allostery; Protein Translocation; SecYEG; SecA; Single-Molecule FRET
**Subject Category** Membranes & Trafficking

## Introduction

A fascinating class of biological molecular machines are those operating upon biopolymer substrates, converting chemical energy derived from ATP binding and hydrolysis into cycles of conformational changes and mechanical work. Examples of these molecular motors include the helicases, unfoldases, chromatin remodelling complexes, primary membrane transporters, protein degradation assemblies (Fairman-Williams et al, 2010; Reyes et al, 2021; Sweeney and Holzbaur, 2018), and the subject of this paper—the protein translocases, exemplified by the ubiquitous secretory (Sec) machinery

The bacterial translocon is minimally composed of the integral inner-membrane core-complex SecYEG, and a peripherally associated cytosolic motor ATPase, SecA (Fig. 1 and Appendix Fig. S1). The complex of the two (SecYEG:A) is necessary and sufficient for the translocation of unfolded polypeptides across membranes in vitro (Brundage et al, 1990; Arkowitz et al, 1993). Transport substrates of this Sec machinery—periplasmic and outer membrane proteins (OMPs)—are transported post-translationally through SecYEG:A as pre-proteins with N-terminal cleavable signal sequences (SS) (Arkowitz et al, 1993; Hartl et al, 1990). SecYEG also directly associates with ribosomes and mediates co-translational insertion of inner membrane proteins (Beck et al, 2000). The mechanism by which SecYEG adapts and performs these various tasks remains unresolved (Allen et al, 2016; Bauer et al, 2014; Catipovic et al, 2019; Collinson, 2019). These core reactions are facilitated by ancillary factors to improve the efficiency of secretion and insertion (Duong and Wickner, 1997; Schulze et al, 2014; Troman et al, 2023; Jauss et al, 2019; Miyazaki et al, 2022).

The protein channel is formed through the centre of SecY (Ma et al, 2019), adjacent to a lateral gate (LG) between transmembrane helices TM2 and TM7, which opens to the bilayer (Van den Berg et al, 2004) (Fig. 1); opening of which is required for pre-protein translocation (du Plessis et al, 2009; Kater et al, 2019). Protein transport is driven by the associated SecA, a DEAD-box ATPase with two nucleotide binding domains, which together form a single ATP binding site (Hunt et al, 2002). The two-helix finger (2HF) domain of SecA has been proposed to act as a sensor regulating nucleotide exchange (Allen et al, 2016; Whitehouse et al, 2012), or alternatively, to directly push the translocating polypeptide across the membrane (Bauer et al, 2014; Catipovic et al, 2019; Catipovic and Rapoport, 2020).

The precise nature of the protein translocation mechanism through SecYEG:A has divided opinion, owing to the inherent

[1]Astbury Centre for Structural Molecular Biology, School of Molecular and Cellular Biology, Faculty of Biological Sciences, University of Leeds, Leeds LS2 9JT, UK. [2]Faculty of Science, University of South Bohemia, České Budějovice 370 05, Czech Republic. [3]School of Clinical and Applied Sciences, Leeds Beckett University, Leeds LS1 3HE, UK. [4]School of Biochemistry, University of Bristol, Bristol BS8 1QU, UK. ✉E-mail: ian.collinson@bristol.ac.uk; fessl@prf.jcu.cz

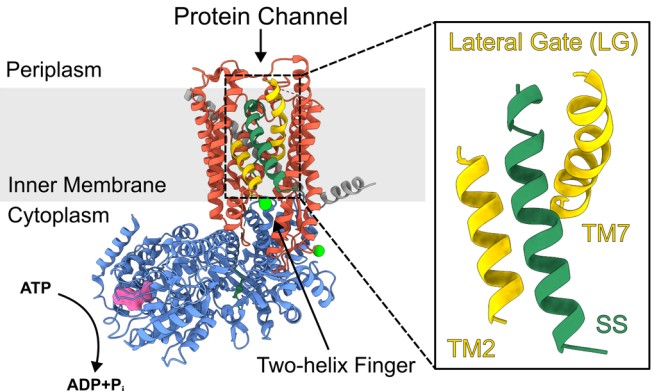

**Figure 1. Electron cryo-microscopy structure of the SecYE:A translocon.**

SecY is shown in red, SecE in dark grey and SecA in blue. The two transmembrane helices that comprise the lateral gate (LG) are shown in yellow (TM2 + TM7) (PDB: 6ITC (Ma et al, 2019), structure determined in the absence of SecG). The inner membrane is shown in light grey. The signal sequence (SS) is shown in dark green. ADP is shown in surface representation (pink) within the nucleotide binding pocket (more details of SecA structure are shown in Appendix Fig. S1). The positions of the mutations used to attach the fluorescent labels used in this work are shown as bright green spheres (A103C and V353C). Note that dye attachment at each site is stochastic with the labelling protocol used in this study.

complexity of the system (Allen et al, 2016; Bauer et al, 2014; Catipovic et al, 2019; Collinson, 2019). As for other molecular machines that convert the chemical potential of nucleoside triphosphate hydrolysis into directional motion, there are two limiting cases of energy transduction: power-stroke and Brownian ratchet (Wagoner and Dill, 2016). The former involves deterministic or direct coupling, in which each conformation imposed by the stage of the nucleotide hydrolysis cycle is linked with a well-defined conformation of the effector part (e.g. a mechanical lever or substrate binding site). On the other hand, a Brownian ratchet mechanism exhibits loose coupling between the conformations of the nucleotide-binding site of the motor ATPase and the effector part (Astumian, 2007). Both cases can be illustrated in terms of a simplified energy profile along the mechanical reaction coordinate (Appendix Fig. S2). The power-stroke profile features deep minima at either the pre-stroke or post-stroke position which directly correspond to the nucleotide state of the ATPase. In the Brownian ratchet case, the nucleotide hydrolysis cycle biases a shallow energy profile (i.e. with low energy barriers between states) towards certain conformations, which in turn undergo rapid interconversion between the available states due to thermal fluctuations.

Power-stroke mechanisms have been demonstrated for many cytoskeletal motors (Lin et al, 2014), while Brownian ratchet schemes have been implicated for other ATP driven systems, such as the ClpX polypeptide unfoldase (Bell et al, 2018). Both mechanisms have been proposed for the SecYEG:A translocon: the power stroke model invokes a large, piston-like motion of the 2HF in SecA, imparting force directly to the substrate, with SecYEG considered a passive pore (Bauer et al, 2014; Catipovic et al, 2019; Catipovic and Rapoport, 2020). Alternatively, SecYEG has been proposed to support a Brownian ratchet, allosterically communicating with SecA (Allen et al, 2016). A recent single-molecule Förster resonance energy transfer (smFRET) study detected

conformational changes of the 2HF taking place on timescales (measured as dwell times in different states) of 100–400 ms (Catipovic et al, 2019). This is similar to the timescale of the ATP hydrolysis cycle (Robson et al, 2009; Allen et al, 2020) (~100 ms) and was interpreted as evidence that SecA acts via a directly coupled power stroke. However, in all published structures of SecYEG:A complexes obtained to date (Zimmer et al, 2008; Park et al, 2014; Li et al, 2016; Ma et al, 2019; Dong et al, 2023) the 2HF is in roughly the same location and there is less than 1 nm available for its movement, unlikely to be enough for the power stroke as proposed in the listed papers above. Moreover, covalent cross-linking of the 2HF to SecYEG does not prevent translocation activity (Whitehouse et al, 2012); emphasising that the 2HF does not need to move much during protein transport and, consequently, any hypothetical power stroke mechanism would either involve other domains of the protein or the step is much smaller than previously proposed.

A recent structural study of SecYE (structures determined in the absence of SecG) and SecA found no movement of the 2HF between two nucleotide occupancy states (ADP·BeF$_3^-$ and ADP) (Dong et al, 2023). Based on these static structures, the authors proposed that two SecA loops in the pre-protein cross-linking domain (PPXD) and the nucleotide binding domains (Appendix Fig. S1) respond to ATP binding and hydrolysis in a fashion similar to monomeric RecA-like helicases and effect directional motion towards the SecYEG channel, once again considered as a passive pore (Dong et al, 2023). However, evidence suggests that SecYEG is far from a static bystander. Indeed, it is well known that the SecYEG channel is actively gated by SecA: biochemical data and molecular dynamics simulations show that the opening and closure of the LG and central channel is linked to the nucleotide state of the associated SecA (Allen et al, 2016; Ahdash et al, 2019). Consistent with this, nucleotide turnover of SecA is affected by SecYEG: (i) the SecA ATP hydrolysis rate increases ~27-fold when associated with SecYEG, and ~760-fold during translocation (Robson et al, 2009); and (ii) ADP release is affected by the channel (and LG) conformation (Allen et al, 2016). Further evidence for two-way allosteric coupling between the channel and motor components was provided by hydrogen-deuterium exchange experiments (Ahdash et al, 2019).

A better understanding of the coupling between SecA and SecYEG is key to the reconciliation of these contrasting mechanisms. High resolution static structures can give detailed information on specific conformations; however, they give no information about the time scales at which these states interchange, and may miss fleetingly populated states. Yet, given the dynamic nature of SecYEG (Mercier et al, 2021), it is likely that transitions between states hold the key to understanding the mechanism of protein transport. To address this gap in our understanding, we have employed smFRET analysis of the dynamic motions of the core SecYEG channel throughout the SecA ATPase cycle. Using multi-parameter photon-by-photon hidden Markov modelling (mpH²MM) (Harris et al, 2022), we provide quantitative analysis of dwell times in the different states observed, and show that SecYEG undergoes transitions between open and closed states on a millisecond time scale—much faster than SecA-catalysed rate of ATP hydrolysis (>100 ms) (Robson et al, 2009; Catipovic et al, 2019). Despite this disparity, the rates of these transitions are controlled by the nucleotide occupancy of SecA. Hence, rather than

a direct coupling of SecA ATP binding/hydrolysis to SecYEG channel motions, the SecA ATPase cycle regulates SecYEG opening and closing by modulating its underlying shallow energy landscape. Quantitative analysis of the state dynamics also revealed a previously undetected, post-hydrolysis, committed state – linked to protein translocation, but preceding the equilibrium ADP-bound state. Finally, we show that rapid translocation by the poorly selective SecY variant PrlA4 (Appendix Fig. S3) correlates with perturbed channel motion, consistent with SecYEG dynamics being a key factor in the process of polypeptide transport.

## Results

### Intrinsic dynamics of SecYEG observed through interconversion between open and closed conformations

To monitor the conformation of the SecYEG channel we employed a previously engineered version of SecY (SecY$_{A103C-V353C}$EG) which contains two unique cysteines introduced into TM2 and TM7 (Fig. 1). This pair, when labelled with fluorescent dyes, was shown to be responsive to SecA and its nucleotide occupancy: a high FRET state (assigned to a closed channel) predominates in the presence of ADP, and a lower FRET state (assigned as an open channel) becomes populated upon addition of the ATP analogue AMP-PNP. The corresponding distances closely match the results of molecular dynamics simulations of the SecYEG:A complex bound respectively to ATP or ADP, supporting these assignments (Allen et al, 2016). We also identified a third FRET signal, arising either from a genuine third state (partially open), or from an average of fast interconversion of the open and closed states. At the time, these possibilities could not be resolved with the camera-based single molecule setup (200 ms per frame) (Allen et al, 2016).

Here, we deployed a single-molecule diffusion-based confocal setup to study dual-labelled SecY$_{A103C-V353C}$EG reconstituted into proteoliposomes comprised of *E. coli* polar lipids (Fig. 2A; Methods), allowing for the detection of conformational dynamics on the sub-millisecond timescale (Agam et al, 2023). The implementation of pulsed interleaved excitation (Müller et al, 2005) provides access to E$_{raw}$ (raw FRET efficiency) and S$_{raw}$ (raw stoichiometry) enabling the use of mpH$^2$MM (Harris et al, 2022) for the detection of transitions and states, and for extraction of the corresponding dwell times (Fig. 2B,C; Appendix Materials and Methods and Appendix Table S1). mpH$^2$MM also effectively deals with contributions from molecules with incomplete fluorophore labelling or those undergoing undesirable photophysics, which can interfere with the analysis of dynamic populations (e.g. donor-only or acceptor-only species and dark donor/acceptor species arising as a result of fluorophore 'blinking').

The analysis for dual labelled SecYEG alone (i.e. in the absence of SecA), in liposomes reconstituted from *E. coli* polar lipid, is described in Fig. 3. mpH$^2$MM followed by the modified Bayes Information Criterion (BIC') analysis (see Appendix Materials and Methods) indicated that the datasets are best explained by four classes (Fig. 3A). Two of these are FRET states associated with conformations of SecYEG, designated as open and closed. This is based on respective low and high FRET efficiency, consistent with the known distances between the two labelled sites in representative crystal structures of the two states (Fig. 3B,C). The remaining two

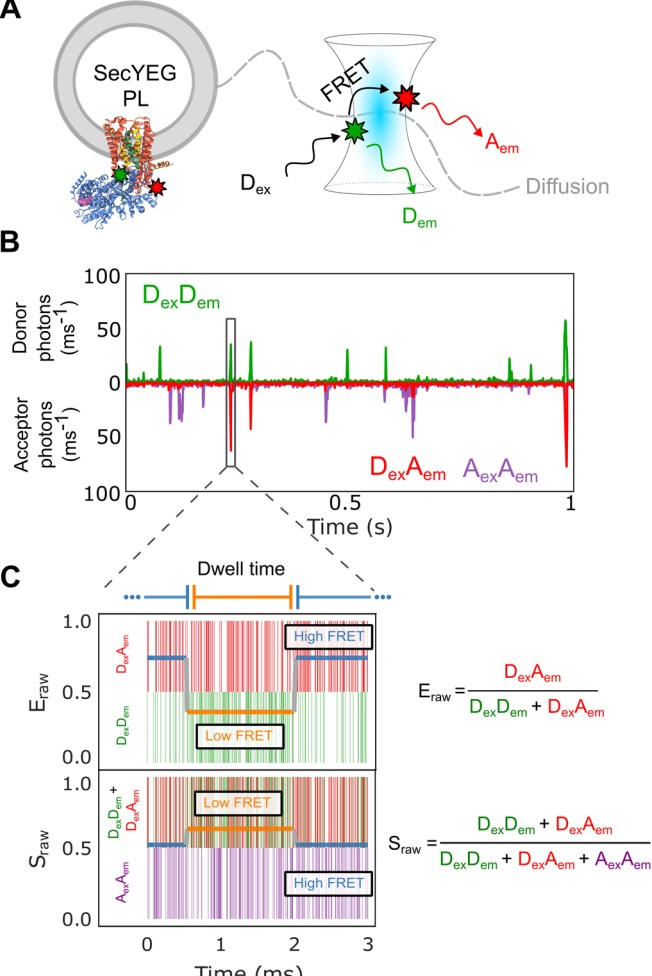

**Figure 2. Single-molecule FRET methodology.**

(A) Confocal volume with illustration of a single proteoliposome (PL) with embedded SecYEG diffusing in and out of the confocal volume alongside. The positions of the dyes are indicted using green/red stars. Note that SecYEG is shown in a single orientation here, but inserts equally in both orientations into the liposomes under the conditions used (Deville et al, 2011). Hence only ~50% of molecules will bind SecA in the experiments presented (see Methods). (B) A representative single-molecule time trace showing photons from the D$_{ex}$D$_{em}$ (Donor Excitation, Donor Emission) in green, D$_{ex}$A$_{em}$ (Donor Excitation, Acceptor Emission) in red, and A$_{ex}$A$_{em}$ (Acceptor Excitation, Acceptor Emission) in purple. (C) A single-molecule photon time trace of a single burst. Photon timestamps are represented by coloured vertical bars (not related to the y-axis). The most likely state path as identified by the Viterbi algorithm as derived by mpH$^2$MM is overlaid as two horizontal coloured lines relating to the y-axis for both E$_{raw}$ and S$_{raw}$.

classes represent contributions from the donor only (or dark acceptor) and acceptor only (or dark donor) species and are therefore not physiologically relevant.

The dynamic nature of the equilibrium between the closed and open states is manifested as transitions on the millisecond time scale (detected by mpH$^2$MM analysis, Fig. 2C and Appendix Materials and Methods) and confirmed by burst variance analysis (BVA) (Torella et al, 2011) (Appendix Materials and Methods and Appendix Fig S5E). Hence, even in the absence of SecA or translocating protein, the SecYEG core-complex is

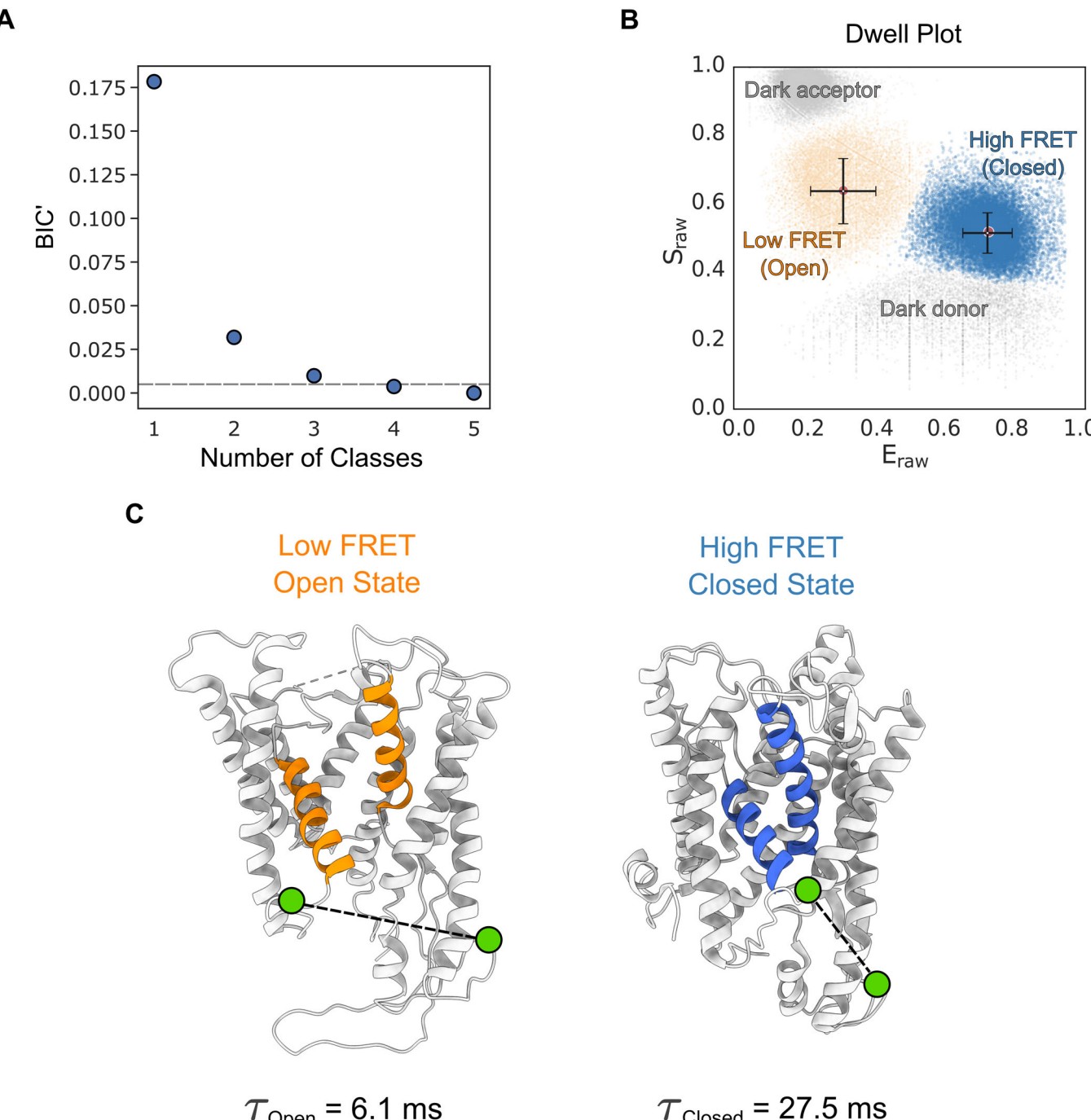

**Figure 3.  Conformation and dynamics of SecYEG protein channel and the adjacent LG are discernible by smFRET on the millisecond timescale.**

(**A**) The modified Bayes Information Criterion (BIC') derived by mpH$^2$MM analysis of SecYEG alone in liposomes. The different state-models indicate that four classes best describe the data (i.e. BIC' < 0.005, grey dashed line). (**B**) Scatterplot of dwells for SecYEG alone reconstituted into proteoliposomes showing the dwell $E_{raw}$ and $S_{raw}$ derived from the mpH$^2$MM analysis. The black bars represent the $E_{raw}$ and $S_{raw}$ standard deviations for each respective population. See Appendix Fig. S5 for more detailed raw data. (**C**) Models highlighting the SecY LG (and channel) in the open (low FRET) and closed (high FRET) conformations (PDBs: Open = 3DIN (Zimmer et al, 2008), Closed = 1RHZ (Van den Berg et al, 2004)). The two fluorophore labelling positions (A103C and V353C on SecY, shown as green circles and connected with a black dashed line) in the open and closed state are indicated. The dwell time ($\tau_{open}$ & $\tau_{closed}$) in each state in SecYEG alone is shown below each figure part. Source data are available online for this figure.

conformationally dynamic, switching between open and closed states on the millisecond timescale.

Previous studies indicated that the conformation of SecYEG is controlled by the nucleotide occupancy of SecA (Allen et al, 2016; Fessl et al, 2018; Ahdash et al, 2019; Catipovic et al, 2019). The complex is also known to respond to the presence of a translocating pre-protein substrate (du Plessis et al, 2009; Fessl et al, 2018; Allen et al, 2016; Ahdash et al, 2019), or to the addition of an isolated SS (Fessl et al, 2018; Gouridis et al, 2009; Hizlan et al, 2012). We therefore collected single-molecule FRET data under a range of different conditions (±SecA, various nucleotides and a SS peptide) for mpH²MM analysis. In each case, the data (Appendix Figs. S5–12) are best described by the four classes (Fig. 3A) assigned to the two conformational states (open and closed) and the irrelevant photophysical donor only and acceptor only events (Fig. 3B). While the two states of interest are always present, their relative proportions change considerably. Furthermore, there is observable rapid interchange on a millisecond timescale between the low and high FRET states (BVA; Appendix Figs. S5e–12e). Thus, each nucleotide state of SecA does not produce a discrete, fixed conformation of SecY, as one might expect based purely on structural data. Instead, SecYEG:A channel behaviour is better described as a dynamic equilibrium of two structural states, open and closed, interconverting on the millisecond timescale.

## SecA and its associated nucleotide modulate the dynamic properties of the protein channel within SecYEG

The mpH²MM methodology allows extraction of the dwell times of the two structural states of SecYEG ($\tau_{open}$ and $\tau_{closed}$). These two parameters contain information not only on the equilibrium position of the two states, but also how fast they interchange. Values for $\tau_{open}$ and $\tau_{closed}$ can readily be visualised in two-dimensional plots (Fig. 4A,B); these reveal the dynamic information obscured in the simple equilibrium view (Fig. 4C), and can thus be used to discern new functional states. The addition of SS to SecYEG (in the absence of SecA), which becomes wedged into the open lateral gate (Ma et al, 2019; Li et al, 2016; Hizlan et al, 2012) (Fig. 1), causes 'unlocking' of the LG, priming SecYEG for protein transport (Corey et al, 2019). Within the dynamic equilibrium context, SS binding dramatically increases the total time SecYEG spends in the open state, not by increasing $\tau_{open}$, but by decreasing $\tau_{closed}$ (from $27.5 \pm 4.6$ ms to $1.3 \pm 0.6$ ms) (Fig. 4A). This equates to a ~4-fold increase in the proportion of the open state (Fig. 4C and Appendix Table S2). By contrast, the addition of SecA (in the absence of nucleotide or SS) tips the balance towards the closed state ~2-fold, by decreasing $\tau_{open}$, with little effect on $\tau_{closed}$ (Fig. 4A,C).

Channel dynamics within the SecYEG:A complex respond to the nucleotide bound to SecA, even in the absence of a pre-protein substrate (Fig. 4B,C). In the presence of SecA there are three distinct types of dynamic behaviour: (i) mostly closed ($95 \pm 4\%$) and relatively static in the SecYEG:A:ADP complex; $\tau_{open}$ is short ($0.5 \pm 0.3$ ms) compared with the long-lived $\tau_{closed}$ ($14.5 \pm 9.4$ ms)); (ii) highly dynamic, but still biased towards closed ($71 \pm 16\%$) seen in the SecYEG:A:ATPγS (a slowly hydrolysing ATP analogue) complex ($\tau_{open}$ is unaffected compared with SecYEG:A:ADP ($0.4 \pm 0.3$ ms) but $\tau_{closed}$ is dramatically decreased ($1.1 \pm 0.3$ ms)); and (iii) moderately dynamic with a larger time spent in the open state ($62 \pm 12\%$) found in the SecYEG:A:ADP·AlF$_x$ complex, which is thought to resemble a post-hydrolysis (PH) transition state

(Goody and Eckstein, 1971; Lacabanne et al, 2020) ($\tau_{open}$ is increased ($3.5 \pm 1.3$ ms) and $\tau_{closed}$ is marginally affected ($1.9 \pm 0.2$ ms)). We also tested the effect of AMP-PNP, a non-hydrolysable analogue of ATP, which as expected elicited the same dynamics in the SecY channel as ATPγS (Appendix Figs. S13 and S14 and Appendix Table 1).

The rate limiting step in the SecA ATPase cycle is ADP release (Robson et al, 2009). Hence, in the absence of pre-protein, the complex spends most of its time (~99%) bound to ADP. We thus expected channel dynamics in the SecYEG:A:ATP steady-state complex (formed by including 1 mM ATP in the experiment ($K_{M[ATP]} = 51.1 \pm 7.8\,\mu M$ (Robson et al, 2009))) to resemble the dynamic position displayed by the SecYEG:A:ADP complex. Remarkably, however, the dynamic equilibrium position of SecYEG:A:ATP is significantly different both to the SecYEG:A:ADP and SecYEG:A:ATPγS complexes (Fig. 4B). Indeed, it does not fall anywhere between these two states, and thus cannot be accounted for by dynamic averaging between the two. Therefore, a third state is required to account for this distinct dynamic behaviour of the channel.

The additional state is unlikely to be the nucleotide-free state, as this is expected to be very transient at 1 mM ATP (Robson et al, 2009), and in any case the dynamic signature is very different from the SecYEG:A complex (see 2D graph, Fig. 4A). Similarly, it is unlikely to be caused by rapid SecA dissociation: again, the dynamic signature does not match the apo form (Fig. 4A), and if SecA could dissociate and rebind on the sub-ms timescale this would most likely also be occurring in the SecYEG:A:ADP complex, and thus already be reflected in the data. Instead, SecYEG:A:ATP dynamics most closely resembles that of SecYEG:A:ADP·AlF$_x$, which is generally reported as a post-hydrolysis state. In this new state, ATP must already have been hydrolysed (otherwise the complex would exhibit dynamics similar to the ATPγS state) and P$_i$ released (P$_i$ release is very fast; $\geq 11.5 \pm 0.07$ s$^{-1}$ compared with a $k_{cat}$ of $0.15 \pm 0.02$ s$^{-1}$ (Robson et al, 2009)). Therefore, SecA must be in an ADP bound conformational state that can only be achieved through hydrolysis of ATP. We designate this state as the post-hydrolysis (PH) form (named SecYEG:A:ADP$^{PH}$) to distinguish it from the predominantly closed complex with loosely (L) associated ADP (SecYEG:A:ADP$^L$), formed by the simple addition and equilibration of ADP with SecYEG:A. This newly assigned PH-state of SecA within the complex clearly imparts distinct dynamics to the SecYEG channel and hence is relevant to the functional cycle of the SecYEG:A.

## Impact of a pre-protein client on the dynamic behaviour of the protein channel

We next examined how the dynamic equilibrium between the open and closed states responds to the translocation of a pre-protein (spheroplast protein Y; proSpy). Spy is a 15.9 kDa periplasmic chaperone (Quan et al, 2011; Kwon et al, 2010), which has been used previously as a model protein for translocation through SecYEG (Allen et al, 2020, 2022; Pereira et al, 2018). The rate of ATP hydrolysis by SecA during translocation of proSpy increases >40-fold (one ATP hydrolysed every 0.15 s compared with one ATP every 6.66 s in the absence of proSpy (Appendix Table S3)). This increase in the observed rate of ATP hydrolysis in the presence of pre-protein is caused by an increased rate of ADP release, increasing the fraction of time spent by the complex in the

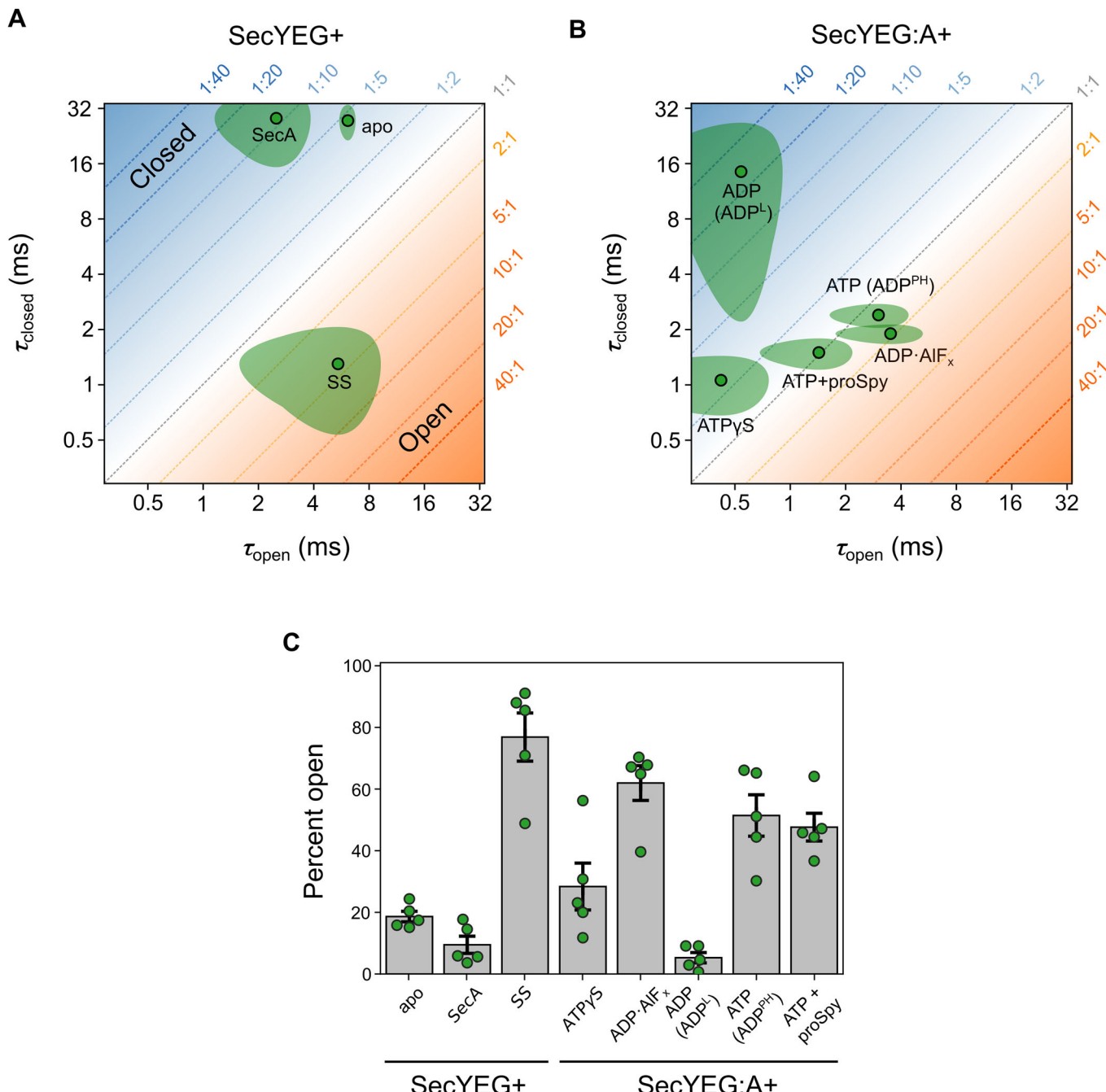

**Figure 4. Dynamics of the SecYEG protein channel and LG in proteolipsomes.**

(A) SecYEG channel dwell times in the open ($\tau_{open}$) and closed ($\tau_{closed}$) states for SecYEG with the addition of (SecYEG+): nothing (apo), SecA or SS. The ratio of time spent in each conformational state is shown as a diagonal dotted line with the associated fraction in each state (open:closed) respectively in orange/blue in the plot margins. (B) As in panel (A), but with the addition of various nucleotides and pre-protein to the SecYEG:A complex (SecYEG:A+). (C) The percent open (i.e. $\tau_{open}/(\tau_{open}+\tau_{closed})\times100$) for data shown in panels (A) and (B). Data information: In (A–C), data is made up from 5 technical repeats (Methods). In (A–B) the mean is shown as a point with a black outline with the 90% confidence interval shown as the surrounding shaded area. In (C) the bars represent the mean value with error bars showing the 90% confidence interval. Raw data for dwell times and percent open are shown in Appendix Tables S1 and S2. smFRET raw data are shown in Appendix Figs. S5–12. Source data are available online for this figure.

ATP-bound pre-hydrolysis state from an average of ~1% to ~37% (Robson et al, 2009). Consistent with this, the steady state dynamic equilibrium of the SecYEG:A:ATP+proSpy complex no longer coincides with the post-hydrolysis ADP state (Fig. 4B; SecYEG:A:ADP$^{PH}$). Instead, it maps to an area between SecYEG:A:ADP$^{PH}$ and the pre-hydrolysis SecYEG:A:ATPγS state (Fig. 4B); thus approximating a steady-state mixture of pre- and post- ATP hydrolysis states.

Taken together, the smFRET data show that the slow ATPase cycle of SecA, which occurs on a >100 millisecond timescale (Robson et al, 2009; Catipovic et al, 2019), modulates intrinsically fast (millisecond timescale) dynamics of the SecYEG core-complex. Each different nucleotide occupancy of SecA produces a distinct pattern of channel opening and closure in SecYEG (Fig. 4B). With ATP but no pre-protein present, the channel spends most of its time in the newly identified ADP$^{PH}$ state, characterised by fast closing and opening. This state is distinct from that induced by the addition of ADP (designated ADP$^L$), in which SecY is predominantly closed. Because the ADP$^{PH}$ state is the main species occupied, we conclude that the rate-limiting step in the ATPase cycle must be the transition from ADP$^{PH}$ to ADP$^L$. This is then followed by very rapid ADP dissociation (hence the designation of ADP$^L$, for loose ADP binding), then immediate binding of another ATP, with ATP hydrolysis returning the complex almost immediately to the ADP$^{PH}$ state.

During translocation, i.e., with a pre-protein present, conversion of ADP$^{PH}$ to ADP$^L$ is accelerated such that it is no longer entirely rate-limiting. ADP dissociation from ADP$^L$ and the subsequent ATP binding remain relatively very fast; however, ATP hydrolysis is now comparable in time scale. The net effect is that the average channel behaviour falls in between pre-hydrolysis (ATPγS; very fast opening and closing) and post-hydrolysis (ADP$^{PH}$; slower but still fast closing and opening). In addition, the total time required for an ATP turnover cycle is much shorter, producing a faster ATPase rate (>40-fold, Appendix Table S3), as expected on the basis of steady state kinetic data (Robson et al, 2009).

## SecYEG channel and lateral gate dynamics are linked to the rate of protein translocation

Next, we explored the role of the rapid millisecond SecYEG dynamics, and its control by the ATPase cycle of SecA, for protein transport. To do so we utilised the widely studied variant of SecYEG PrlA4, which contains two amino acid substitutions in SecY, F286Y and I408N (in TM7 & TM10 respectively (Appendix Fig. S3)) (Nouwen et al, 1996; Emr et al, 1981). This variant was produced by an *E. coli* strain selected to suppress the effects of a defective signal sequence. This suppression is achieved by the resulting SecYEG-PrlA4 complex being primed in an 'unlocked' conformation, which would otherwise require the docking of a functional SS at the LG (Emr et al, 1981; Bondar et al, 2010; Osborne and Silhavy, 1993; Van den Berg et al, 2004; Corey et al, 2016). Given that SS insertion dramatically affects channel dynamics (Fig. 4A), we hypothesised that the PrlA4 variant may also exhibit perturbed dynamics such that the open state is promoted.

Experiments measuring the rate of translocation of proSpy (see Methods) showed that the PrlA4 variant has a translocation rate ~3.4-fold greater (9.70 ± 0.02 aa.s$^{-1}$) than that of regular SecYEG

(2.85 ± 0.03 aa.s$^{-1}$) (Fig. 5A and Appendix Tables S4 and S5), while having approximately the same ATP hydrolysis rate by SecA during translocation (Fig. 5B and Appendix Table S3). These data reveal two important points: (i) the SecYEG-PrlA4 variant is more efficient at translocation of pre-protein substrates (Fig. 5C, 1.38 ± 0.09 aa/ATP for PrlA4 compared to unaffected SecYEG 0.44 ± 0.03 aa/ATP) and (ii) the ATP hydrolysis cycle seems not to be strictly correlated with the rate of proSpy translocation, which would be expected if SecA was acting alone as a stepping motor.

To determine how the channel dynamics are affected in the PrlA4 variant the two amino acid substitutions F286Y and I408N were incorporated into SecYEG along with A103C and V353C for labelling and proteoliposome reconstitution, exactly as described above for the unsubstituted control (see Methods). Channel variant dynamic behaviour was again analysed by smFRET. The results revealed that PrlA4 exhibits the same two FRET states observed in the regular SecYEG (Appendix Figs. S15–18), corresponding to the same open and closed conformations. However, the interconversion between these states is drastically different (Appendix Table S6). In the apo complex (in the absence of SecA/nucleotides/SS) PrlA4 is more dynamic than SecYEG ($\tau_{open}$ = 1.5 ± 0.3 ms and $\tau_{closed}$ = 4.5 ± 1.8 ms (Fig. 5D) for PrlA4, compared with $\tau_{open}$ = 6.1 ± 0.5 ms and $\tau_{closed}$ = 27.5 ± 4.6 ms for SecYEG (Fig. 4A)). While more mobile, the overall percentage of time spent in the open state is roughly the same (PrlA4 = 26 ± 4% compared to control SecYEG = 19 ± 4%, Appendix Tables S2 and S7).

Upon the addition of SecA, the dynamic equilibrium of PrlA4 is shifted significantly towards the open configuration (60 ± 6%), with a rate of dynamic interchange similar to that observed for the SecYEG standard bound to the SS peptide ($\tau_{open}$ = 2.4 ± 0.6 ms and $\tau_{closed}$ = 1.6 ± 0.2 ms) (Fig. 5D). Therefore, perhaps unsurprisingly, PrlA4 can be unlocked by SecA (more open) without the need for a SS, explaining its ability to translocate pre-proteins with defective signal-sequences. Consistent with this the unlocked PrlA4-SecA complex (without SS) is indistinguishable from PrlA4-SS (Fig. 5D). Previous reports suggested that the PrlA4 variant cause a general 'relaxation' in SecYEG, rather than a specific conformational change which allows for the bypassing of signal sequence recognition (Nouwen et al, 1996). Our findings support this notion, but additionally show that association of SecA is necessary and sufficient for the complex to adopt the unlocked state.

During translocation, SecYEG-PrlA4:A:ATP+proSpy alternates between longer dwells in both the open and closed channel states ($\tau_{open}$ = 4.5 ± 1.5 ms $\tau_{closed}$ = 8.3 ± 3.3 ms), in contrast to the control complex (SecYEG:A:ATP+proSpy), which is more dynamic under the same conditions ($\tau_{open}$ = 1.4 ± 0.6 ms and $\tau_{closed}$ = 1.4 ± 0.5 ms) (Fig. 5E). Interestingly, the total percentage of time spent in the open state by PrlA4 during translocation (36 ± 10%) is similar to that of normally behaving SecYEG (48 ± 9%) (Appendix Tables S2 and S7). Hence, the equilibrium position between open and closed states is not responsible for the phenotype of *prlA4*. Rather, it seems to be the time the channel spends in the open state ($\tau_{open}$) that accounts for the faster transport phenotype. Consistent with this, the increase in the dwell time of the open conformation in PrlA4 (~3.1-fold compared with SecYEG; Fig. 5E) corresponds closely to the observed increase in protein translocation rate (~3.4-fold; Fig. 5A,E).

Passive polypeptide diffusion has been shown to contribute significantly to the kinetics of translocation (Bauer et al, 2014), and passage of positively charged and bulky regions across the membrane is

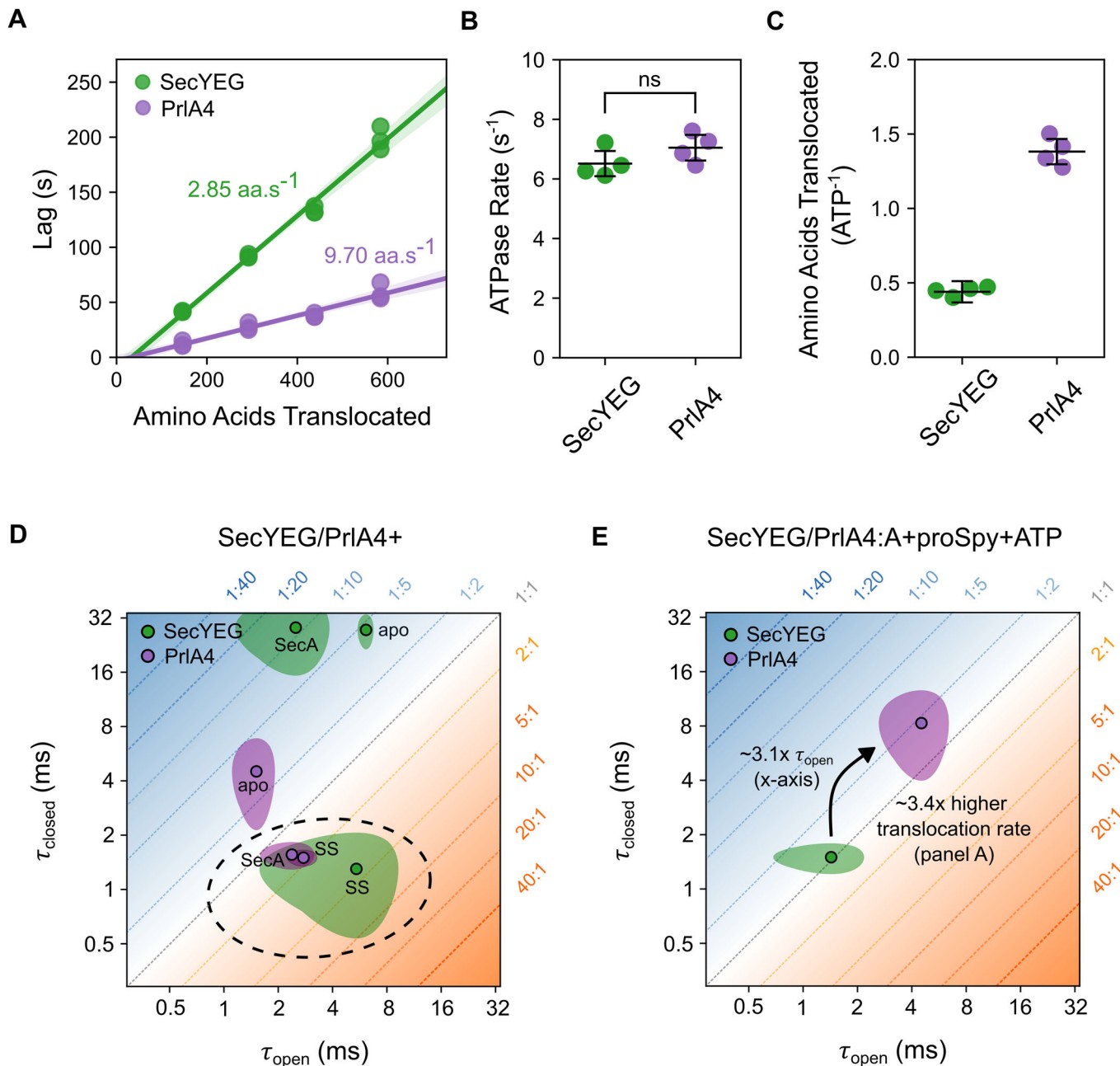

**Figure 5. The SecYEG-PrlA4 variant enables more efficient protein translocation compared to the native SecYEG complex.**

(A) Rates of transport of proSpy for SecYEG and SecYEG-PrlA4 as measured by the NanoLuc split-luciferase assay. The assay measures the time lag before the onset of chemiluminescence due to binding of high affinity β-strand, which is inserted at various positions into a 4x tandem repeat of proSpy, to NanoLuc luciferase fragment encapsulated within SecYEG proteoliposomes ($n = 3$ technical repeats for each proSpy variant, Methods and Appendix Materials and Methods, data shown in Appendix Tables S4 and S5). A linear regression fit is shown as a solid line, with the 95% confidence interval shown (95%CI) shaded in the respective colour. (B) Steady-state rates of ATP hydrolysis by SecA translocating proSpy through SecYEG vs SecYEG-PrlA4 (ns, $p$-value $= 0.18$, $T$-test) ($n = 4$ technical repeats). (C) The number of amino acids translocated by SecYEG per ATP hydrolysed by SecA during transport of proSpy through regular SecYEG ($0.44 \pm 0.03$) and PrlA4 ($1.38 \pm 0.09$) (calculated from the respective rate of transport derived in panel (A) and the steady-state rates of ATP hydrolysis in panel (B). (D) LG dwell times of SecYEG-PrlA4, SecYEG-PrlA4:A and SecYEG-PrlA4:SS in proteoliposomes (purple) compared with the control SecYEG complex (green, data for the latter are reproduced from Fig. 4A for comparison). Conditions representing the 'unlocked' state of the channel (i.e. primed for transport and have an equilibrium position favouring the open state) are circled with a grey dashed line. (E) Channel dwell times of the translocation of proSpy through PrlA4, coloured as in (D) with the steady-state dwell time in the open state and translocation rates compared with that of unaffected SecYEG. Data information: In (D, E), data is made up from 5 technical repeats (Methods) and shown as the mean (point with black outline) with 90% confidence interval (shaded area). The error bars in (B, C) are given as the 95% confidence interval. Source data are available online for this figure.

rate limiting for the overall transport process (Allen et al, 2022). The simplest explanation for the observed PrlA4 effect, therefore, is that a long dwell time in the open state is required for such regions to cross the membrane. Normally, the channel through SecYEG only very rarely stays open long enough for a full transport event to occur; for PrlA4, such long opening events are much more frequent. Thus, the altered translocation properties of the PrlA4 variant are not due to SecA motor action, but instead are an effect of the duration of the open state of the protein channel in SecYEG. In this scenario the increased dwell time of the closed state is inconsequential.

## Discussion

### Energy landscape steering of SecYEG protein channel dynamics

In this study we have exploited the powers of smFRET analyses to show that rapid millisecond dynamic interchange between two conformational states is an intrinsic property of SecYEG, independent of SecA. The closed state is the most highly populated (~81%) for SecYEG alone at equilibrium. Binding of SecA (without nucleotide) to SecYEG shifts the equilibrium to further promote channel closure (to >91%) by decreasing the dwell time of the open state. This allosteric effect is further modulated by the nucleotide state of SecA.

Based on previous biophysical studies from our own group and elsewhere (Bauer et al, 2014; Catipovic et al, 2019; Catipovic and Rapoport, 2020; Dong et al, 2023; Allen et al, 2016; Corey et al, 2019; Ahdash et al, 2019; Fessl et al, 2018; Allen et al, 2020, 2022; Li et al, 2016) it was anticipated that the conformations of the SecYEG protein channel would mirror the slow (>100 ms) ATPase cycle of the associated SecA. In other words, discrete states of the channel would exist that are strictly constrained by the nucleotide bound to SecA—interconverting upon ATP hydrolysis and ADP exchange for ATP. As we show here, it is only in the SecYEG:A:ADP$^L$ complex that the channel may be considered quasi-static (~95% closed), and this state is only appreciably populated in the non-physiological condition where ADP is present in the absence of ATP. Binding of ATP, or non-hydrolysable analogues, to SecA shifts the population of the dynamic equilibrium towards the open state. This is achieved by inducing fast, millisecond time scale interconversion between the open and closed states. The SecA nucleotide dependent control over rapid SecYEG channel and LG dynamics can be visualised with the help of simplified energy landscapes projected onto a reaction coordinate of opening and closure (Fig. 6A). Within this framework SecA 'steers' the energy landscape by changing the relative depth of the two minima (equilibrium) and the barrier height between them (interconversion rate).

To better visualise the relationship between the hydrolytic cycle of SecA and the dynamics of the SecYEG channel, we projected a representative interconversion trajectory of the open and closed states onto an 'average' ATPase cycle (during translocation; Fig. 6B). The time SecA spends associated with each nucleotide was determined using two measured parameters: (i) the steady-state ATPase rate (Fig. 5B), and (ii) the rate of phosphate release (Robson et al, 2009). We further assumed that ATP binding at the millimolar concentration used is much faster than any subsequent steps (Robson et al, 2009; Natale et al, 2004), and thus the apo SecYEG:A (without nucleotide) does not contribute to the cycle.

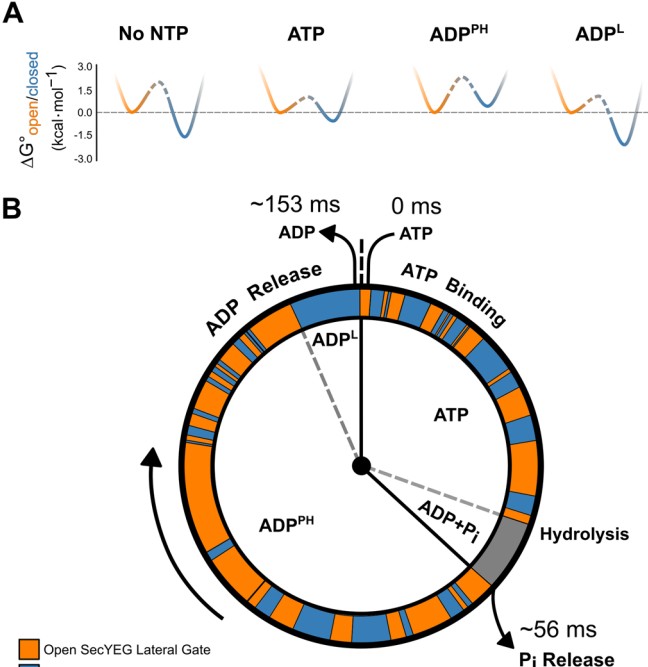

Figure 6. Observed SecYEG protein channel dynamics as a function of the ATP hydrolysis cycle of SecA.

(A) Free energies between conformational states of the SecYEG were estimated from data for the following conditions: No NTP = SecYEG:A, ATP = SecYEG:A:ATPγS, ADP$^{PH}$ = SecYEG:A:ATP, ADP$^L$ = SecYEG:A:ADP. The closed state well is coloured blue while the open state is orange. Barrier heights are difficult to compute quantitatively and are therefore represented in dashed lines (see Appendix section 'Calculation of simplified potential energy surface' for more details). (B) The ATP hydrolysis cycle of SecA is shown as a 'clock' with a simulated trajectory of channel conformations shown in orange (open) and blue (closed) around the perimeter. The representative trajectory was simulated with the experimentally derived rates of interconversion. The time spent in each step during translocation is calculated from the $k_{cat}$ of ATP hydrolysis ($6.52 \pm 0.42\ s^{-1}$) (Fig. 5B) and the time taken for phosphate release ($17.9 \pm 0.15\ s^{-1}$) (Robson et al, 2009).

Likewise, since phosphate release is fast (Robson et al, 2009), we assume the system spends a relatively short time in the ADP+P$_i$ post-hydrolysis state. Currently, we do not know if the transient pre-phosphate release step is more like the ATP or ADP$^{PH}$ bound complexes, or something in between. During translocation, the steady-state mixture of nucleotide-associated SecYEG:A complexes can be best approximated by a mixture of the ATP-bound (ATPγS) and ADP$^{PH}$ (ATP) forms. Hence, we estimated that the ADP$^{PH}$ form of the complex to dominate the ADP associated phase of the cycle, with the contribution of the ADP$^L$ (loosely associated) form being marginal.

The kinetic cycle we describe highlights the stark difference in timescales between SecYEG channel dynamics and those of ATP hydrolysis by SecA. The results clearly demonstrate that SecYEG and SecA cannot be directly coupled, and that thermal fluctuations of the channel and the adjacent LG play an important role in the pre-protein translocation mechanism. These data support previously proposed models suggesting a dynamic equilibrium between open and closed conformations of SecYEG based on lateral gating being strongly dependent on temperature (Ge et al, 2014). Thus, the functional state of the SecYEG:A complex cannot

be represented by a single structure. Instead, protein translocation will involve an ensemble of co-existing conformations interchanging on a rapid timescale; effectively in a dynamic equilibrium controlled by the SecA ATPase cycle. Thereby, SecA 'steers' the conformational ensemble through a sequence of different dynamic equilibria by modulating the landscape according to its nucleotide state (Fig. 6).

Previous electrophysiology experiments have demonstrated that apo SecYEG maintains a membrane seal by action of the plug, preventing passage of small ions and water through the channel (Saparov et al, 2007). In agreement with this our previous results show that the plug only fully opens in the presence of SecA, ATP and translocating substrate (Fessl et al, 2018). Furthermore, inside the cell the apo-state is likely to be rare: the translocon will be mostly be occupied with SecA and translating ribosomes on the cytosolic surface, as well as chaperones and quality factors on the periplasmic side, providing additional insulation for the channel. Therefore, the action of the plug and extrinsic binding partners explain how the secretion machinery prevents unwanted dissipation of electrochemical potential; mitigating transient opening of the isolated SecYEG reported here.

## Fast protein channel and LG dynamics are integral to the mechanism of translocation

Previous cross-linking experiments demonstrated that channel and LG opening is essential for translocation (du Plessis et al, 2009) and that closure slows down ADP release (the rate limiting step of ATP turnover) from SecA (Allen et al, 2016). In contrast, translocation models based solely on SecA motor action predict the translocation speed to be directly linked to the ATP hydrolysis rate, independent of channel dynamics (Catipovic et al, 2019; Dong et al, 2023). Here, by smFRET analysis, we show unequivocally that the dynamics of the SecYEG channel are an integral part of the translocation mechanism.

We found that the SecY-PrlA4 variant exhibits a faster (~3.4 times more rapid) pre-protein translocation rate at the same ATP cost. Importantly, we determine that the site-specific changes in PrlA4 do not affect the equilibrium position of open versus closed channel, but significantly increase the dwell times of both states. This increase in dwell time (~3.1 times slower interconversion) correlates well with the increased rate of translocation (~3.4 times faster). The longer dwell time of the open state would increase the probability of charged and bulky residues within the pre-protein sequence being translocated through the channel. This observation might also explain the reported ability of the PrlA4 variant to translocate partially folded pre-protein substrates (Nouwen et al, 1996).

We show here that SecYEG operates on distinct timescales to the ATP hydrolytic cycle (Robson et al, 2009) and associated conformational dynamics of SecA; the latter also being shown by single-molecule analysis (Catipovic et al, 2019; Chada et al, 2018). The evident stochastic coupling of separate events at the channel and the ATPase are not compatible with a directly coupled power stroke. The evolution of such a mechanism with interacting, but dynamically independent, components could be versatile and widespread. The shallow energy landscape of the different states of the core SecYEG complex enables rapid access to different conformational states and interconversion between them. The 'steering' of these energy landscapes, in conjunction with accessory

factors, could then be exploited for different activities, such as insertion of trans-membrane helices into the bilayer. While the results here reveal the mismatch in timescales in the Sec translocon, the precise mechanism by which the 'slow' SecA ATP binding/hydrolysis cycle allosterically modulates the energy landscape of the 'fast' SecY still is not clear. To better understand this coupling (especially regarding conformational changes within the complex by which ADP release can occur) would require multi-colour smFRET methods spanning multiple timescales (μs–s) to monitor conformational changes in SecA and SecY simultaneously.

Such a dynamic allosteric mechanism is not unique to the SecYEG translocon, since similar nucleotide-dependent steering of rapid dynamics has also been recently shown for adenylate kinase (Aviram et al, 2018) and the AAA+ ring ATPase, ClpB (Mazal et al, 2019). These insights were also revealed by advances in single-molecule fluorescence techniques. Other ATP-driven pumps also have proposed mechanisms incorporating a series of static states associated with or without different nucleotides (e.g. the ABC transporters (Thomas and Tampé, 2020)). In view of the work described here it would not be surprising if rapid conformational interconversion between key transport states, i.e., inward and outward facing, are also subject to the influence of the various stages of the ATP hydrolytic cycle, operating at a different timescale. It may indeed be the case that dynamic allostery involving motions occurring across multiple time scales coupled via energy landscape steering is ubiquitous among complex molecular machines.

# Methods

## Protein expression, purification and labelling

Protein production, purification and labelling were performed according to protocols published previously (Allen et al, 2016, 2020; Collinson et al, 2001; Gold et al, 2007). *E. coli* SecYEG with two unique Cysteine (Cys, C) residues, A103C and V353C, in SecY (SecY$_{A103C-V353C}$EG) was used and labelled with ATTO 565 as a donor dye and ATTO 643 as an acceptor dye for single-molecule FRET measurements, as previously described (Allen et al, 2016). The PrlA4 variant (Nouwen et al, 1996; Osborne and Silhavy, 1993), which also contained the amino acid substitutions A103C and V353C, was created by site-directed mutagenesis before being purified and labelled as described for the regular version. Note that under the conditions used (100 μM SecY$_{A103C-V353C}$EG, 200 μM each of ATTO 565 maleimide and ATTO 643 maleimide), labelling of each Cys with each fluorophore is random. The translocation precursor, proSpy, which contains Spy with its natural signal sequence, was expressed in an *E. coli* strain defective in the export of secreted proteins and purified to homogeneity as described by Pereira et al (Pereira et al, 2018). Full details can be found in the Appendix Materials and Methods.

## ATPase assays

ATPase assays were performed and analysed as described by Gold et al (Gold et al, 2007), modified to allow data collection on a BioTek Synergy Neo2 plate reader; full details can be found in the Appendix Materials and Methods.

## Transport assays

Transport assays were performed as described by Pereira et al (Pereira et al, 2018), and the data analysed as in Allen et al (Allen et al, 2020). Full details can be found in the Appendix Materials and Methods.

## SecYEG proteoliposome preparation

SecYEG proteoliposomes were prepared for single-molecule FRET as in Allen et al (Allen et al, 2016) with minor adjustments: labelled SecY$_{A103C-V353C}$EG was reconstituted to a concentration of 37.5 nM into *E. coli* polar lipid extract (10060 C, Avanti Polar Lipids) at a volume of 400 µl and a concentration of 5 mg/ml in TKM buffer (20 mM Tris, 50 mM KCl, 2 mM MgCl$_2$, pH 7.5) to achieve a protein to lipid ratio of 1:106666. The mixture was then extruded to form proteoliposomes with a diameter of 100 nm. With this liposome size and protein to lipid ratio the liposomes are expected to contain either 0 or 1 copy of SecY$_{A103C-V353C}$EG (ratios of 1:73000 have been shown to produce ~95% singly occupied or unoccupied liposomes) (Deville et al, 2011). The extrusion step was performed on a heating block set to 40 °C (610000-1EA, Avanti Polar Lipids). The resulting mixture was dialysed overnight using D-Tube Dialyzer Mini with a Molecular weight cut-off of 12–14 kDa (Sigma Aldrich) in TKM buffer before being stored at 4 °C.

## Single-molecule data acquisition

Single-molecule Förster resonance energy transfer (smFRET) experiments were performed on a custom-built confocal epi-illuminated microscope (Appendix Materials and Methods) in a standard inverted-stage configuration with a pulsed interleaved excitation regime (Müller et al, 2005). Samples were measured in an 8-well sample chamber (80827, Ibidi) which had been coated in bovine serum albumin (BSA) to prevent any adhesion of the sample to the chamber. Coating of the sample chamber was achieved by pipetting 500 µl of 1 mg/ml BSA solution which had been filtered through a 0.22 µm membrane and leaving at room temperature for 20 min. After this time, the BSA solution was removed and the sample chamber was rinsed thoroughly with MilliQ water before being left to air dry. Samples were measured at a concentration of 30 pM in TKM buffer supplemented with 1 mM aged Trolox to help reduce blinking and photobleaching of the fluorophores. Trolox was aged to form a fraction of oxidised Trolox (Trolox-quinone) which then acts with Trolox according to a reducing and oxidizing system scheme (Vogelsang et al, 2008; Cordes et al, 2009). Aged Trolox was made by adding 1 mM of Trolox to TKM buffer and left overnight on a shaker at 4 °C to dissolve before being filtered using a 0.22 µm membrane. The chamber was covered during data acquisition to help prevent evaporation of the sample. Relevant components for each condition were added immediately before measurement to final concentrations of 1 µM SecA, 1 µM SS, 0.7 µM proSpy and 1 mM ATP, ATPγS, ADP and ADP·AlF$_x$. ATP depletion was negligible due to low SecYEG:A concentration and low intrinsic turnover of free SecA (Robson et al, 2009). Each condition was measured 5 times for 1 h (apart from ATP +proSpy, where each measurement for 20 min) with fresh proteoliposome preparations. Appendix Fig. S19 shows the variability in the data of the five fresh proteoliposome preparations for SecYEG apo.

## Single-molecule data analysis

Data were analysed using the FRETBursts python package (Ingargiola et al, 2016). A burst search with a minimum threshold of 6x the background signal in the donor and acceptor channels respectively and a minimum burst size of 50 photons was used to distinguish single molecule events. Further details on burst selection, mpH$^2$MM and statistical analyses are given in the Appendix Materials and Methods. Due to the random nature of the insertion of SecY$_{A103C-V353C}$EG into the liposomes, ~50% of the protein was oriented in such a way that the SecA binding interface was inaccessible (Mao et al, 2013; Schulze et al, 2014; Allen et al, 2016), therefore we applied a correction factor to account for the remaining non-participating 50% apo (Appendix Fig S4, Appendix Materials and Methods).

# Data availability

The smFRET data associated with this paper are openly available from the University of Leeds Data Repository (https://doi.org/10.5518/1410).

# Peer review information

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

## Acknowledgements

This work was funded by the BBSRC: JAC, RT and SER (BB/T008059/1); WJA and IC (BB/V001531/1); DWW and IC (BB/T006889/1). SER holds a Royal Society Professorial Research Fellowship (RSRP\R1\211057). JAC, TF and RT are supported by the European Regional Development Fund-Project (CZ.02.1.01/0.0/0.0/15_003/0000441) and the Czech Science Foundation (20-11563Y). JAC and TS also acknowledge Leeds Beckett University for funding. We thank David Brockwell for his feedback and discussion and G. Nasir Khan for his excellent technical support. We also thank Matthew Watson for performing data collection in the early stages of the project.

## Author contributions

**Joel A Crossley**: Conceptualization; Data curation; Software; Formal analysis; Validation; Investigation; Methodology; Writing—original draft; Writing—review and editing. **William J Allen**: Conceptualization; Formal analysis; Validation; Investigation; Methodology; Writing—original draft; Writing—review and editing. **Daniel W Watkins**: Conceptualization; Investigation; Visualization; Methodology. **Tara Sabir**: Supervision; Funding acquisition; Methodology. **Sheena E Radford**: Conceptualization; Resources; Supervision; Funding acquisition; Investigation; Writing—original draft; Project administration; Writing—review and editing. **Roman Tuma**: Conceptualization; Resources; Supervision; Funding acquisition; Investigation; Writing—original draft; Project administration; Writing—review and editing. **Ian Collinson**: Conceptualization; Resources; Supervision; Funding acquisition; Investigation; Writing—original draft; Project administration; Writing—review and editing. **Tomas Fessl**: Conceptualization; Resources; Data curation; Software; Formal analysis; Supervision; Funding acquisition; Validation; Investigation; Visualization; Methodology; Writing—original draft; Project administration; Writing—review and editing.

## Disclosure and competing interests statement

The authors declare no competing interests.

