## [Peer Review File · The EMBO Journal]

Dynamic coupling of fast channel gating with slow ATP-turnover underpins protein transport through the Sec translocon

Ian Collinson, Joel Crossley, William Allen, Daniel Watkins, Tara Sabir, Sheena Radford, Roman Tuma, and Tomas Fessl
DOI: [10.15252/emboj.2023114897](https://doi.org/10.15252/emboj.2023114897)

Corresponding author(s): Ian Collinson (ian.collinson@bristol.ac.uk) , Tomas Fessl (tomfessl@gmail.com)

Review Timeline:

Submission Date:	30th Jun 23
Editorial Decision:	2nd Aug 23
Revision Received:	6th Sep 23
Editorial Decision:	28th Sep 23
Revision Received:	9th Oct 23
Editorial Decision:	3rd Nov 23
Revision Received:	6th Nov 23
Accepted:	8th Nov 23

Editor: William Teale

Transaction Report:

Dear Dr. Collinson,

Thank you again for the submission of your manuscript entitled "Energy landscape steering mediates dynamic coupling in ATP-driven protein translocation" (EMBOJ-2023-114897). We have now received reports from three referees, which I copy below.

As you can see from their comments, whilst referee 1 raises technical concerns that will need to be carefully considered, referees 2 and 3 are clearly supportive of publication.

Based on the overall interest expressed in these reports, I would like to invite you to address the comments of all referees in a revised version of the manuscript. I should add that it is The EMBO Journal policy to allow only a single major round of revision and that it is therefore important to resolve the main concerns at this stage. I believe the concerns of the referees are reasonable and addressable, but please contact me if you have any questions, need further input on the referee comments or if you anticipate any problems in addressing any of their points. I am available for a Zoom call next week if you would like to discuss referee 1's report, this may be particularly useful on the issue of directly measuring SecA-dependent ATP hydrolysis. Please, follow the instructions below when preparing your manuscript for resubmission.

I would also like to point out that as a matter of policy, competing manuscripts published during this period will not be taken into consideration in our assessment of the novelty presented by your study ("scooping" protection). We have extended this 'scooping protection policy' beyond the usual 3 month revision timeline to cover the period required for a full revision to address the essential experimental issues. Please contact me if you see a paper with related content published elsewhere to discuss the appropriate course of action.

Again, please contact me at any time during revision if you need any help or have further questions.

Thank you very much again for the opportunity to consider your work for publication. I look forward to your revision.

Best regards,

William

William Teale, Ph.D.
Editor
The EMBO Journal

When submitting your revised manuscript, please carefully review the instructions below and include the following items:

- 1) a .docx formatted version of the manuscript text (including legends for main figures, EV figures and tables). Please make sure that the changes are highlighted to be clearly visible.
- 2) individual production quality figure files as .eps, .tif, .jpg (one file per figure).
- 3) a .docx formatted letter INCLUDING the reviewers' reports and your detailed point-by-point response to their comments. As part of the EMBO Press transparent editorial process, the point-by-point response is part of the Review Process File (RPF), which will be published alongside your paper.
- 4) a complete author checklist, which you can download from our author guidelines ([https://wol-prod-cdn.literatumonline.com/pb-assets/embo-site/Author Checklist%20-%20EMBO%20J-1561436015657.xlsx](https://wol-prod-cdn.literatumonline.com/pb-assets/embo-site/Author%20Checklist%20-%20EMBO%20J-1561436015657.xlsx)). Please insert information in the checklist that is also reflected in the manuscript. The completed author checklist will also be part of the RPF.
- 5) Please note that all corresponding authors are required to supply an ORCID ID for their name upon submission of a revised manuscript.
- 6) We require a 'Data Availability' section after the Materials and Methods. Before submitting your revision, primary datasets produced in this study need to be deposited in an appropriate public database, and the accession numbers and database listed under 'Data Availability'. Please remember to provide a reviewer password if the datasets are not yet public (see <https://www.embopress.org/page/journal/14602075/authorguide#datadeposition>). If no data deposition in external databases is

needed for this paper, please then state in this section: This study includes no data deposited in external repositories. Note that the Data Availability Section is restricted to new primary data that are part of this study.

Note - All links should resolve to a page where the data can be accessed.

8) For data quantification: please specify the name of the statistical test used to generate error bars and P values, the number (n) of independent experiments (specify technical or biological replicates) underlying each data point and the test used to calculate p-values in each figure legend. The figure legends should contain a basic description of n, P and the test applied. Graphs must include a description of the bars and the error bars (s.d., s.e.m.).

9) We would also encourage you to include the source data for figure panels that show essential data. Numerical data can be provided as individual .xls or .csv files (including a tab describing the data). For 'blots' or microscopy, uncropped images should be submitted (using a zip archive or a single pdf per main figure if multiple images need to be supplied for one panel). Additional information on source data and instruction on how to label the files are available at .

10) We replaced Supplementary Information with Expanded View (EV) Figures and Tables that are collapsible/expandable online (see examples in <https://www.embopress.org/doi/10.15252/embj.201695874>). A maximum of 5 EV Figures can be typeset. EV Figures should be cited as 'Figure EV1, Figure EV2" etc. in the text and their respective legends should be included in the main text after the legends of regular figures.

12) Our journal encourages inclusion of *data citations in the reference list* to directly cite datasets that were re-used and obtained from public databases. Data citations in the article text are distinct from normal bibliographical citations and should directly link to the database records from which the data can be accessed. In the main text, data citations are formatted as follows: "Data ref: Smith et al, 2001" or "Data ref: NCBI Sequence Read Archive PRJNA342805, 2017". In the Reference list, data citations must be labeled with "[DATASET]". A data reference must provide the database name, accession number/identifiers and a resolvable link to the landing page from which the data can be accessed at the end of the reference. Further instructions are available at .

Further instructions for preparing your revised manuscript:

We realize that it is difficult to revise to a specific deadline. In the interest of protecting the conceptual advance provided by the work, we recommend a revision within 3 months (31st Oct 2023). Please discuss the revision progress ahead of this time with the editor if you require more time to complete the revisions. Use the link below to submit your revision:

Referee #1:

The Sec translocon is the major route for protein export across the cytoplasmic membrane of prokaryotes (and essential the same translocon is responsible for protein entry into the eukaryotic endoplasmic reticulum). Thus, detailed mechanistic studies of the operation of the Sec translocon are of great importance for cellular biology.

In bacteria protein transport through the Sec translocon is normally post-translational and powered by the ATP-driven 'motor' SecA. In this study the authors have used cutting edge single molecule FRET measurements to characterize the kinetics of opening of the Sec translocon and how these are modulated by SecA during different stages in the cycle of ATP hydrolysis and substrate engagement. From these measurements the authors provide a convincing case that the role of ATP hydrolysis is to enable SecA to bias the opening of the Sec channel to allow diffusive movement of the substrate protein. This contrasts with the majority view that the substrate protein is mechanically pushed in a power stroke mechanism. This conclusion is a paradigm shift in our understanding of Sec mechanism. The work will be of great significance for the protein transport field and more generally for the understanding of biological machines.

The work is extremely well presented and argued, given the complexity of the methods. The technical nature of the work is outstanding and thorough. My comments on the manuscript are mainly presentational.

This study relies on the assignment of the two FRET states that are identified and analyzed as corresponding to the physiological open and closed states of the translocon. The evidence for this assignment is given as (Allen et al, 2016). Whilst I agree that the assignment is highly plausible and that the data in Allen et al. is consistent with this interpretation, that paper does not seem to report FRET measurements from states unambiguously trapped in the two states. For this reason, I would recommend rather than just citing Allen et al. that the authors summarize the key reasons why that paper supports the assigned states.

It is absolutely vital to the interpretation of the data that the reader knows that the SecYEG complexes have been reconstituted into proteoliposomes. I didn't initially pick up that this was what the authors had done even though it is mentioned where the main text talks about labelling and shown in Fig. 2A. It might be worthwhile to state in the figure legends that the samples being analysed are proteoliposomes. Part of my confusion was that the methods describe isolation of SecYEG in detergent but as far as I can see do not describe reconstitution into proteoliposomes.

If the Sec translocon is 20% open without partners (apo state) as the authors observe how does the cell maintain the membrane seal? The authors should also note and comment on previous electrophysiology data (Saparov et al. (2007) *Mol Cell* 26: 501-509) that indicated that the apo state does not gate open and that the PrIA4 variant shows opening for extended periods of time.

Pg 2. 'The complex of the two (SecYEG:A) is necessary and sufficient for the translocation of unfolded polypeptides across the inner membrane (Brundage et al, 1990; Arkowitz et al, 1993).' It would be advisable to qualify this as 'in vitro' as there are studies that claim that PMF is essential for transport in vivo.

Pg 4. `...there is less than 1 nm available for 2HF movement, not enough for the proposed power stroke.' It is not obvious to me why a movement of this scale would not be enough for a powerstroke. Either give the more detailed rationale or qualify as `unlikely to be enough'.

Pg 11.

`Thus, the rate limiting step is a conformational change that involves the SecYEG channel and adjacent LG and is associated with the conversion of SecYEG:A:ADPPH to a state from which ADP is readily released, most likely the equilibrium SecYEG:A:ADPL state.'

This needs to be rewritten for clarity because I'm not sure I have followed what is being argued here. I can see that the rate limiting step is as described in the second half of the sentence but what is the link to SecYEG conformational change? If this is that the two states have different SecYEG opening regimes then this connection should be made.

Fig. 2B. Even if conventional in the field, the presentation here is difficult to understand as it looks as though the burst are higher and lower than a baseline whereas the two halves of the graph refer to bursts at two different emission wavelengths (I assume). So either label the two halves on the y-axis as being associated with different wavelengths and/or have the acceptor emission trace as a replica panel going from 0 to 1 rather than from 1 to 0.

Similarly, in 2C it needs to be clearer in the legend and figure labelling that the y-axis scale is plotting the running average/modelled FRET states and that the vertical lines are just time stamps for photons and not related to the y-axis quantities.

Fig 5d. It would be preferable to label in this figure which is the variant data and which the wild type rather than relying on carrying over colours from other panels. What are the grey dotted lines? Are they necessary? Should they be arrows?

Fig 5e. What do the comments written on the panel mean? Topen is presumably from the data shown but is the efficiency of translocation is from elsewhere and so can't be calculated from this graph? This should be made clear.

Fig 6a. It should be made clear that the y-axis scale is proportional and assigned from quantitative calculations (rather than being illustrative) but that the scale is relative not absolute. Maybe have numbers on the axis and label these as relative delta G? Although the calculation is given in the SI methods the nature of the calculation in the figure legend is too minimal. What is the nature of the `data' that are going into the calculation to produce state thermodynamics? It was not clear to me what had been done even from the methods.

Referee #2:

The precision of protein transport by the Sec translocon is vital for all aspects of life. However, the underlying molecular mechanism remain elusive. In this manuscript, Crossley et al. investigated the fundamental principle of protein translocation using single molecule FRET and biochemical reconstitution. The results depicted a comprehensive framework of the translocon dynamics during the ATP hydrolysis cycle of SecA, demonstrating protein translocation occurs mostly likely through the Brownian ratchet model instead of the power stroke model as proposed by previous studies based on ensemble biochemical measurements. Therefore, the findings reported by Crossley et al. resolve a long-standing debate about the Sec translocon and provide novel insights into our understanding of protein translocation. The results are very exciting and will become a 'must-read' paper for people interested in single molecule biophysics. The manuscript is well-written with self-explained figures. I am highly supportive in the publication of the manuscript if a few minor issues could be clarified as detailed below.

1. One potential issue is that all smFRET measurements were performed in proteoliposomes in which 50% of SecY will not be available for engaging SecA or substrate. To address this problem, the authors carried out extensive simulations to determine the "corrected rates". However, these data were not included in the paper. As such, it is challenging to understand the processing of the smFRET data. It would be very useful if the authors could provide these simulations and also a more detailed procedure in the analysis of their smFRET data.
2. The other issue of using proteoliposomes is the copy number of the Sec translocon. For single molecule imaging experiments, it is critical to toss out oligomers from monomers. However, I am not able to figure out how the authors know that the proteoliposomes analyzed in this study only contain monomers.
3. Since the authors have determined the rate constants in the conformational transitions of the Sec translocon, it should be possible to calculate the energy of each state and include those values in the free energy profiles.

Referee #3:

Crossley et al. reported the opening dynamics of the SecYEG channel in different translocation states using smFRET. SecYEG is one of the most conserved and ubiquitous channels (known as Sec61 in eukaryotes) responsible for translocating secretory proteins across lipid membranes. In bacteria, the channel is coupled with the ribosome for co-translational translocation of nascent peptide chains, and with SecA for post-translational translocation of peptide chains. SecA is a highly conserved ATPase that provides energy for protein translocation. The translocation mechanism of SecA with SecYEG has been studied for decades, with two major working models currently under debate: the Brownian ratchet model and the power-stroke model. The authors' group has published a series of papers supporting the Brownian ratchet model, and they have employed smFRET with similar experimental settings in their previous work (Allen et al., *elife* 2016).

This manuscript presents smFRET data from faster recording and an improved fitting model. With these improvements, the authors could better define the open and close frequencies at the lateral gate of the channel in different nucleotide and SecA binding states. The different timescales observed between the dynamics of the SecY gate and SecA hydrolysis led the authors to conclude that SecA modulates SecYEG conformation to facilitate protein translocation. While the refined measurements of SecYEG dynamics is informative, the current data is too preliminary to draw the conclusion.

Major points:

1. The estimated timescale of ATP hydrolysis by SecA was based on other biochemical assays. As smFRET is super sensitive to conformational changes, the dynamics of SecA in the similar smFRET settings should be measured as well to justify the conclusion. The authors could label different domains of SecA, measure the internal dynamics, and assess the distance between SecA and SecYEG in different nucleotide and translocation states.
2. The PH state is puzzling. The use of different nucleotides to define the ATP hydrolysis states should be done with caution. For example, in the ADP state, SecA may rapidly alternate between being on and off from the channel. It is unclear whether the reported gate dynamics reflect the average behavior (once again, the dynamics of SecA in this state should be measured). The exact states mimicked by ATP analogs, such as ATP γ S, AMPPNP, ADP.BeFx, and ADP.AlFx, have been interpreted differently in various ATPase systems. Notably, the AMPPNP data was reported in previous work (Allen et al., *elife* 2016), but not here. Any specific reasons? This reviewer suggests including more supporting data to clearly define each state, beyond the apparent gate opening and closing frequencies. Additional data would strengthen the study's conclusions and provide a more comprehensive understanding of the PH state (if exists).
3. The discussion on the rate limiting step in Page 11 lines 3-8 is unclear. Please elaborate how such conclusion is drawn.
4. The authors suggest that the leakage of the Prl4 mutant may be explained by the longer open state. However, the overall translocation rates are supposed to reflect the average of open and closed states. If we consider the ratio of open and closed states, the wild-type (wt) actually appears to be even better than Prl4 (compare figs 4b and 5e).
5. In the last sentence of the abstract, the authors seem to infer that the channel is static according to previous structural studies. However, this is not true. It is well known from structural and biochemical studies that the SecYEG channel is dynamic and can open up to accommodate substrates of different sizes. SecA binding also contributes to opening up and tightening the channel. In the main text, while providing a detailed description of the channel dynamics, the authors should refrain from over-emphasizing the importance of their results by suggesting that the SecA-mediated dynamic nature of the channel was not known from previous structural data.

Response to referees: EMBOJ-2023-114897

“Energy landscape steering mediates dynamic coupling in ATP-driven protein translocation”

Key: black = verbatim referee comment, blue = our response.

Referee #1:

The Sec translocon is the major route for protein export across the cytoplasmic membrane of prokaryotes (and essential the same translocon is responsible for protein entry into the eukaryotic endoplasmic reticulum). Thus, detailed mechanistic studies of the operation of the Sec translocon are of great importance for cellular biology.

In bacteria protein transport through the Sec translocon is normally post-translational and powered by the ATP-driven ‘motor’ SecA. In this study the authors have used cutting edge single molecule FRET measurements to characterize the kinetics of opening of the Sec translocon and how these are modulated by SecA during different stages in the cycle of ATP hydrolysis and substrate engagement. From these measurements the authors provide a convincing case that the role of ATP hydrolysis is to enable SecA to bias the opening of the Sec channel to allow diffusive movement of the substrate protein. This contrasts with the majority view that the substrate protein is mechanically pushed in a power stroke mechanism. This conclusion is a paradigm shift in our understanding of Sec mechanism. The work will be of great significance for the protein transport field and more generally for the understanding of biological machines.

The work is extremely well presented and argued, given the complexity of the methods. The technical nature of the work is outstanding and thorough. My comments on the manuscript are mainly presentational.

This study relies on the assignment of the two FRET states that are identified and analyzed as corresponding to the physiological open and closed states of the translocon. The evidence for this assignment is given as (Allen et al, 2016). Whilst I agree that the assignment is highly plausible and that the data in Allen et al. is consistent with this interpretation, that paper does not seem to report FRET measurements from states unambiguously trapped in the two states. For this reason, I would recommend rather than just citing Allen et al. that the authors summarize the key reasons why that paper supports the assigned states.

R1-A1: We have now expanded the opening of the results section to include a summary of the data in (Allen et al, 2016) that provides evidence for the FRET mutant SecY_{A103C-V353C}EG reporting on the opening and closure of the SecY lateral gate.

It is absolutely vital to the interpretation of the data that the reader knows that the SecYEG complexes have been reconstituted into proteoliposomes. I didn't initially pick up that this was what the authors had done even though it is mentioned where the main text talks about labelling and shown in Fig. 2A. It might be worthwhile to state in the figure legends that the samples being analysed are proteoliposomes. Part of my confusion was that the methods describe isolation of SecYEG in detergent but as far as I can see do not describe reconstitution into proteoliposomes.

R1-A2: We have now made reference to proteoliposomes in the Results section and figure legends. We have also split the proteoliposomes reconstitution protocol into its own section in the Methods.

If the Sec translocon is 20% open without partners (apo state) as the authors observe how does the cell maintain the membrane seal? The authors should also note and comment on previous electrophysiology data (Saparov et al. (2007) Mol Cell 26: 501-509) that indicated that the apo state does not gate open and that the PrIA4 variant shows opening for extended periods of time.

R1-A3: Referee #1 brings up a great point here and highlights the importance of consolidation of our data showing spontaneous opening of channel with previous electrophysiology data showing that apo wild-type SecY is impermeable to ions and water (Saparov et al. 2007). We have added a paragraph to the discussion section explaining how we believe these two findings can be reconciled.

Pg 2. 'The complex of the two (SecYEG:A) is necessary and sufficient for the translocation of unfolded polypeptides across the inner membrane (Brundage et al, 1990; Arkowitz et al, 1993).' It would be advisable to qualify this as 'in vitro' as there are studies that claim that PMF is essential for transport in vivo.

R1-A4: We thank Referee #1 for pointing out this important distinction which we have now added to the manuscript.

Pg 4. '...there is less than 1 nm available for 2HF movement, not enough for the proposed power stroke.' It is not obvious to me why a movement of this scale would not be enough for a powerstroke. Either give the more detailed rationale or qualify as 'unlikely to be enough'.

R1-A5: We have amended the text to state '*unlikely to be enough for the power stroke as proposed in the above papers.*' and we agree with Referee #1 that this is appropriate wording given the statement's subjective nature. We also expanded the end of the paragraph to clarify the reasoning behind our point.

Pg 11. 'Thus, the rate limiting step is a conformational change that involves the SecYEG channel and adjacent LG and is associated with the conversion of SecYEG:A:ADPPH to a state from which ADP is readily released, most likely the equilibrium SecYEG:A:ADPL state.'

This needs to be rewritten for clarity because I'm not sure I have followed what is being argued here. I can see that the rate limiting step is as described in the second half of the sentence but what is the link to SecYEG conformational change? If this is that the two states have different SecYEG opening regimes then this connection should be made.

R1-A6: We agree with Referee #1 that the terminology used here was confusing. We have significantly expanded this section and used terminology which we think makes the text easier to follow.

Fig. 2B. Even if conventional in the field, the presentation here is difficult to understand as it looks as though the burst are higher and lower than a baseline whereas the two halves of the graph refer to bursts at two different emission wavelengths (I assume). So either label the two

halves on the y-axis as being associated with different wavelengths and/or have the acceptor emission trace as a replica panel going from 0 to 1 rather than from 1 to 0.

Similarly, in 2C it needs to be clearer in the legend and figure labelling that the y-axis scale is plotting the running average/modelled FRET states and that the vertical lines are just time stamps for photons and not related to the y-axis quantities.

R1-A7: We thank the reviewer for these suggestions which we have implemented, making the figures easier to understand. The Figures/Legends are amended accordingly.

Fig 5d. It would be preferable to label in this figure which is the variant data and which the wild type rather than relying on carrying over colours from other panels. What are the grey dotted lines? Are they necessary? Should they be arrows?

Fig 5e. What do the comments written on the panel mean? Topen is presumably from the data shown but is the efficiency of translocation is from elsewhere and so can't be calculated from this graph? This should be made clear.

R1-A8: We have added labels to panels Fig 5D and E as suggested. We also opted to remove the grey dashed lines from Fig 5D (originally added to help guide the reader) to avoid any confusion. We have added in parentheses to the annotations in Fig 5E to inform the reader where the data comes from.

Fig 6a. It should be made clear that the y-axis scale is proportional and assigned from quantitative calculations (rather than being illustrative) but that the scale is relative not absolute. Maybe have numbers on the axis and label these as relative delta G? Although the calculation is given in the SI methods the nature of the calculation in the figure legend is too minimal. What is the nature of the 'data' that are going into the calculation to produce state thermodynamics? It was not clear to me what had been done even from the methods.

R1-A9: We have amended Fig 6A to include the changes Referee #1 has suggested. We also changed the introduction to the Appendix section 'Calculation of simplified potential energy surface' to make it clear that the data used was the transition rates.

Referee #2:

The precision of protein transport by the Sec translocon is vital for all aspects of life. However, the underlying molecular mechanism remain elusive. In this manuscript, Crossley et al. investigated the fundamental principle of protein translocation using single molecule FRET and biochemical reconstitution. The results depicted a comprehensive framework of the translocon dynamics during the ATP hydrolysis cycle of SecA, demonstrating protein translocation occurs mostly likely through the Brownian ratchet model instead of the power stroke model as proposed by previous studies based on ensemble biochemical measurements. Therefore, the findings reported by Crossley et al. resolve a long-standing debate about the Sec translocon and provide novel insights into our understanding of protein translocation. The results are very exciting and will become a 'must-read' paper for people interested in single molecule biophysics. The manuscript is well-written with self-explained figures. I am highly supportive in the publication of the manuscript if a few minor issues could be clarified as detailed below.

1. One potential issue is that all smFRET measurements were performed in proteoliposomes in which 50% of SecY will not be available for engaging SecA or substrate. To address this problem, the authors carried out extensive simulations to determine the "corrected rates". However, these data were not included in the paper. As such, it is challenging to understand the processing of the smFRET data. It would be very useful if the authors could provide these simulations and also a more detailed procedure in the analysis of their smFRET data.

R2-A1: We apologise that this was not clear. We have now added a figure outlining our 'Apo correction' process (Appendix Fig S4) and clarified text in the Appendix section 'Correction for the presence of apo SecYEG', which we refer to in the main text.

2. The other issue of using proteoliposomes is the copy number of the Sec translocon. For single molecule imaging experiments, it is critical to toss out oligomers from monomers. However, I am not able to figure out how the authors know that the proteoliposomes analyzed in this study only contain monomers.

R2-A2: We have updated our renamed Materials and Methods section 'SecYEG Proteoliposome Preparation' to better clarify how we prepared proteoliposomes without more than 1 copy of SecYEG.

3. Since the authors have determined the rate constants in the conformational transitions of the Sec translocon, it should be possible to calculate the energy of each state and include those values in the free energy profiles.

R2-A3: We thank Reviewer #3 for the suggestion and we have now updated Fig 6A to reflect the changes suggested (see response R1-A9 above).

Referee #3:

Crossley et al. reported the opening dynamics of the SecYEG channel in different translocation states using smFRET. SecYEG is one of the most conserved and ubiquitous channels (known as Sec61 in eukaryotes) responsible for translocating secretory proteins across lipid membranes. In bacteria, the channel is coupled with the ribosome for co-translational translocation of nascent peptide chains, and with SecA for post-translational translocation of peptide chains. SecA is a highly conserved ATPase that provides energy for protein translocation. The translocation mechanism of SecA with SecYEG has been studied for decades, with two major working models currently under debate: the Brownian ratchet model and the power-stroke model. The authors' group has published a series of papers supporting the Brownian ratchet model, and they have employed smFRET with similar experimental settings in their previous work (Allen et al., *elife* 2016).

This manuscript presents smFRET data from faster recording and an improved fitting model. With these improvements, the authors could better define the open and close frequencies at the lateral gate of the channel in different nucleotide and SecA binding states. The different timescales observed between the dynamics of the SecY gate and SecA hydrolysis led the authors to conclude that SecA modulates SecYEG conformation to facilitate protein

translocation. While the refined measurements of SecYEG dynamics is informative, the current data is too preliminary to draw the conclusion.

Major points:

1. The estimated timescale of ATP hydrolysis by SecA was based on other biochemical assays. As smFRET is super sensitive to conformational changes, the dynamics of SecA in the similar smFRET settings should be measured as well to justify the conclusion. The authors could label different domains of SecA, measure the internal dynamics, and assess the distance between SecA and SecYEG in different nucleotide and translocation states.

R3-A1: We agree with the referee that the dynamic coupling in the Sec translocon needs to be investigated further to all the other factors associated with translocation (SecA and many others) ideally simultaneously with those in SecYEG. However, doing this will be technically very challenging and will form the basis of a whole series of experiments; enough to form another manuscript. This will obviously take some time, beyond the scope of this study, but will hopefully form the basis of our next submission! We have written a section of text summarising the technical challenges in the Discussion section 'Fast protein channel and LG dynamics are integral to the mechanism of translocation' second paragraph.

Overall, however, we do not believe this omission undermines the important conclusions of the manuscript. Previous research by the Rapoport Lab used total internal reflection fluorescence microscopy to measure smFRET on SecA and recover the rates of conformational change during protein transport similar to the way Referee #3 suggests (Catipovic *et al.* 2019). Importantly, the results showed that the rates matched well to the ensemble ATP hydrolysis rates measured by their group and are similar to the values which we present here, suggesting that the method of measurement has little quantitative effect on the determined values.

Furthermore, we would note that the important new conclusions of our paper do not rest on the exact rate constants within the ATPase cycle. The evidence is very strong for the mismatched timescales between the conformational dynamics of SecYEG and SecA, and for the assignment of different conformational equilibria to specific nucleotide states. It is upon these robust comparisons that we make our conclusions.

2. The PH state is puzzling. The use of different nucleotides to define the ATP hydrolysis states should be done with caution. For example, in the ADP state, SecA may rapidly alternate between being on and off from the channel. It is unclear whether the reported gate dynamics reflect the average behavior (once again, the dynamics of SecA in this state should be measured). The exact states mimicked by ATP analogs, such as ATP γ S, AMPPNP, ADP.BeFx, and ADP.AIFx, have been interpreted differently in various ATPase systems. Notably, the AMPPNP data was reported in previous work (Allen *et al.*, *elife* 2016), but not here. Any specific reasons? This reviewer suggests including more supporting data to clearly define each state, beyond the apparent gate opening and closing frequencies. Additional data would strengthen the study's conclusions and provide a more comprehensive understanding of the PH state (if exists).

R3-A2: The reviewer is absolutely correct that the exact interpretation of different nucleotide analogues should be treated with caution. Concerning the first part of the question, i.e. the PH state – we consider it highly unlikely that SecA dissociation can explain the SecYEG:A:ATP

state. Firstly, the PH state has a dynamic signature in the 2D graph that is very different from both 'apo' and ADP states. More generally, if SecA were dissociating and rebinding on a timescale fast enough to contribute to a dynamic average (i.e. ms timescale), it would presumably also be doing this with ADP present. If so, this would already be reflected in the SecYEG:A:ADP data, which has clearly different dynamics. Thus, we only invoke the PH state because it seems impossible to explain the SecYEG:A:ATP data in any other way. We have rewritten the relevant section to make this argument clearer.

With regards to the initial choice of ATP γ S over AMPPNP, the reason was two-fold. 1) Research by Fak et al. which showed that AMP-PNP may not serve as faithful mimic of ATP in its interactions with SecA due to its affinity to SecA being at least 100x weaker than ATP γ S (Fak et al. 2004). 2) ATP γ S has been used in numerous studies on SecA-SecY interactions as a non-hydrolysable analogue and we wanted to be able to compare our data with these other reports, most notably smFRET by Catipovic et al. 2019 (important also for response to point 1 above); Bauer et al. 2014; Catipovic and Rapoport 2020. However, we have now also collected data for AMP-PNP (five independent hour long data acquisitions, measured in the same way as all other conditions). The data for ATP γ S and AMP-PNP match well (now shown in new Appendix Figs. S13 and S14 and Appendix Table 1) giving us further confidence that the dynamic ensemble we measured matches the pre-hydrolysis state in the SecYEG:A complex and allows us to compare our present conclusions with our previous TIRF data as well as the more recent work on SecA by others.

3. The discussion on the rate limiting step in Page11 lines 3-8 is unclear. Please elaborate how such conclusion is drawn.

R3-A3: We apologise this was not clear. We have now significantly expanded this section to better explain the conclusion to our findings and think this paragraph is now much better. Thank you for pointing this out.

4. The authors suggest that the leakage of the Prl4 mutant may be explained by the longer open state. However, the overall translocation rates are supposed to reflect the average of open and closed states. If we consider the ratio of open and closed states, the wild-type (wt) actually appears to be even better than Prl4 (compare figs 4b and 5e).

R3-A4: At first, we also expected to see the PrlA4 mutant spending a greater proportion of its time in the open state, to account for the increased transport rate. However, as the reviewer points out, this is not what the data show. Therefore, we must conclude that it is the dwell time in the open state, rather than the ratio of open and closed states, that determines overall transport rate.

Our current working hypothesis is that a successful 'transport event' – i.e. passage of a difficult stretch of pre-protein across the membrane – requires the channel to stay open for a relatively long time (say 4 ms). Therefore, transport time is dictated by the frequency of long-duration opening events, not just the average channel position. This would explain why PrlA4 transports much faster despite spending the same (or more) time in the closed state. We suggest that the longer dwell time in the closed state is inconsequential.

We think this is an important observation, and as the referee points out, is perhaps counter-intuitive. We have amended the section where this is discussed to make it clearer.

5. In the last sentence of the abstract, the authors seem to infer that the channel is static according to previous structural studies. However, this is not true. It is well known from structural and biochemical studies that the SecYEG channel is dynamic and can open up to accommodate substrates of different sizes. SecA binding also contributes to opening up and tightening the channel. In the main text, while providing a detailed description of the channel dynamics, the authors should refrain from over-emphasizing the importance of their results by suggesting that the SecA-mediated dynamic nature of the channel was not known from previous structural data.

R3-A5: We absolutely did not intend to imply that previous studies show a static SecYEG channel. Rather, we were attempting to highlight the importance of rapid dynamics and energy landscape steering for the mechanism of translocation. These ms-timescale dynamics are the major new insight of our manuscript, which have been made possible by recent advances in smFRET methodology and analysis.

We have amended and extended the final section of the abstract to make this clearer, and added better descriptions of the dynamic equilibrium and how it differs from previous results to the main text.

Together, we believe these changes improve the manuscript to better reflect the novelty of our findings. We hope that it is now clear that the manuscript goes beyond the known properties of SecA to alter the conformation of SecY. Rather, the dynamics of the Sec machinery are manifested through multiple timescales orders of magnitude apart, modulated allosterically within SecYEG *via* energy landscape steering. We believe that this concept of dynamic allostery will not only be essential to understand how protein translocation occurs in the Sec machinery, and other systems, but have wider implications in understanding other complex molecular machines through cutting edge single-molecule fluorescence techniques.

References

- Bauer, Benedikt W., Tom Shemesh, Yu Chen, and Tom A. Rapoport. 2014. "A 'Push and Slide' Mechanism Allows Sequence-Insensitive Translocation of Secretory Proteins by the SecA ATPase." *Cell* 157 (6): 1416–29.
- Catipovic, Marco A., Benedikt W. Bauer, Joseph J. Loparo, and Tom A. Rapoport. 2019. "Protein Translocation by the SecA ATPase Occurs by a Power-Stroke Mechanism." *The EMBO Journal*, March, e101140.
- Catipovic, Marco A., and Tom A. Rapoport. 2020. "Protease Protection Assays Show Polypeptide Movement into the SecY Channel by Power Strokes of the SecA ATPase." *EMBO Reports*, September, e50905.
- Deville, Karine, Vicki A. M. Gold, Alice Robson, Sarah Whitehouse, Richard B. Sessions, Stephen A. Baldwin, Sheena E. Radford, and Ian Collinson. 2011. "The Oligomeric State and Arrangement of the Active Bacterial Translocon." *The Journal of Biological Chemistry* 286 (6): 4659–69.
- Fak, John J., Anna Itkin, Daita D. Ciobanu, Edward C. Lin, Xiang-Jin Song, Yi-Te Chou, Lila M. Gierasch, and John F. Hunt. 2004. "Nucleotide Exchange from the High-Affinity ATP-Binding Site in SecA Is the Rate-Limiting Step in the ATPase Cycle of the Soluble Enzyme and Occurs through a Specialized Conformational State." *Biochemistry* 43 (23): 7307–27.
- Saparov, Sapar M., Karl Erlandson, Kurt Cannon, Julia Schaletzky, Sol Schulman, Tom A. Rapoport, and Peter Pohl. 2007. "Determining the Conductance of the SecY Protein Translocation Channel for Small Molecules." *Molecular Cell* 26 (4): 501–9.
- Sobakinskaya, Ekaterina, Heinrich Krobath, Thomas Renger, and Frank Müh. 2021. "Structural Determinants of a Permeation Barrier of the SecYEG Translocon in the Active State." *Physical Chemistry Chemical Physics: PCCP* 23 (45): 25830–40.

Dear Ian,

I have now received comments from all three referees on the revised version of your manuscript. I have included these reports at the bottom of this email. As you will see, referees 1 and 2 are satisfied by the changes that you have made. Unfortunately, referee 3 remains unconvinced that your experimental design is able accurately to test the hypotheses which you present. This report is clearly written and well argued.

The technical nature of referee 3's concerns makes it impossible for me to accept the manuscript for publication in its current form. However, to be completely fair to you and your co-authors, I would like to offer you another opportunity to respond to the points raised and find common ground. Please let me know if you judge that another round of experiments could help, as this possibility is also still open.

Yours sincerely,

William

William Teale, PhD
Editor
The EMBO Journal
w.teale@embojournal.org

Referee #1:

The authors have satisfactorily dealt with all my review comments. From my point of view they have also satisfactorily addressed the other reviewers' comments.

Referee #2:

In this manuscript, Crossley et al. provided novel insights into the molecular mechanism of the protein translocon using state-of-art single molecule approaches. The authors have well addressed my previous concerns. I am delighted to support the publication of this study at EMBO J.

Referee #3:

The revised manuscript has improved clarity in writing. However, a major concern still persists regarding the interpretation of the dynamics of SecY in various nucleotide binding states, particularly with the inclusion of the AMPPNP data in the revision. As previously pointed out in the comments, it is essential to exercise caution when defining ATP hydrolysis states using different nucleotide analogs. In this case, the FRET data indicates that, when combined with SecA, the SecY channel is mostly in the closed state with ADP, which is consistent with other biochemical and structural data. However, the FRET data obtained with different ATP analogs are not consistent with each other.

1. ATP γ S is typically considered a slowly hydrolysable ATP analog, rather than strictly a non-hydrolysable ATP, as stated in the manuscript. In the FRET experiment setting (with recordings lasting ~ 1 hour), it becomes challenging to estimate the extent to which ATP γ S has been converted to ADP. For instance, ATP γ S has been employed in many AAA-ATPase systems to capture subunits in various states, including ATP, ADP, and apo states, for biochemical and structural studies. Thus, it is not convincing to treat the ATP γ S data as the pre-hydrolysis state with the current experimental settings.
2. ADP.AIFx and ADP.BeFx are non-hydrolysable and exhibit a high affinity for SecA-SecY. The addition of these compounds to SecA-SecY can stabilize the complex in an open conformation, as demonstrated in the structure by Zimmer J et al. Nature, 2008. The observed increase in open time in the FRET experiments could be attributed to the open conformation of SecY in the structure.
3. It is not surprising to observe similar LG dynamics in SecA-SecY between ATP binding and ADP.AIFx binding. Rather than proposing a new PH state as suggested by the authors, a more straightforward interpretation could be that ATP binding also keeps the channel open due to the structural similarity between ATP and ADP.AIFx.
4. The AMPPNP data presented in the revised manuscript introduce further confusion. As cited by the authors, Fak JJ et al. reported that AMPPNP exhibited a much lower affinity for SecA compared to ATP γ S, and even lower than ADP. However, the FRET data reported in this study indicate that AMPPNP and ATP γ S resulted in similar SecY dynamics, which is difficult to

reconcile. More notably, the AMPPNP data in this manuscript appear to contradict the findings from an earlier paper by the same group, Allen WJ et al. eLife 2016. In that earlier work, AMPPNP was regarded as a non-hydrolysable ATP analog, and its FRET data showed significant stimulation of SecY channel opening, to an extent even greater than ATP binding (Fig. 5 of the paper).

Considering these conflicting data and interpretations, it is quite challenging to persuade this reviewer of the conclusions drawn regarding the dynamics of the channel during protein translocation solely based on the FRET data and subsequent analyses. Concerning the explanation that Topen of Prl4 dictates the translocation process, do the authors think that a single opening event (e.g., within 4 ms) could allow the passage of an entire protein strand? If not, when a protein peptide is present in the channel without signal sequence peptides (SS), will the FRET signal indicate an open state or a closed state? In other words, do the FRET experiments in this report distinguish between the opening of the lateral gate induced by SS (at the initialization stage, fig.4a), SS and a translocating peptide inside the channel (at the early translocation stage), and a peptide alone in the channel (at the middle and later translocation stages)? The authors do not seem to have considered this difference. However, in the reviewer's opinion, it is crucial to define the relationship between these various "open" states and the FRET signal before proceeding with further analysis. Presumably, an idle channel has a closed lateral gate (LG), an active channel with SS and a translocating peptide has an open LG. An active channel with only a translocating peptide has an LG opening somewhere between these two states. Neglecting to address this issue could potentially affect the interpretation of the Topen/Tclosed plots for the ATP+proSpy sample.

Dr William Teale
Scientific Editor
The EMBO Journal

Ian Collinson
Professor of Biochemistry
School of Biochemistry
University Walk
Bristol BS8 1TD
United Kingdom

Tel.: +44 117 33-12131
Fax: +44 117 33-12168
Mobile: +44 780 95-39701
E-mail: ian.collinson@bristol.ac.uk

5th October, 2023

Dear Dr Teale,

Thank you for sending us the referees' comments on our revised manuscript. We were pleased to see that referees 1 and 2 recommend publication. We are grateful for the opportunity to respond reviewer 3; because, fundamentally, their second round of objections do not stand up to technical scrutiny. Rather, the referee appears to have preconceived opinions about how ATPases ought to work, which are incompatible with our results. As such, more data – however compelling – is unlikely to make any difference to their viewpoint.

The main point of contention is the assignment of the state observed in the presence of ATP γ S and AMPPNP (the latter performed at the referee's suggestion). These are slowly- and non-hydrolysing analogues of ATP (respectively), and are widely used to capture pre-hydrolysis conformations of ATPases. While both differ slightly from ATP, they clearly represent pre-hydrolysis conformations because the terminal phosphate (or thiophosphate) is covalently bonded to the ADP moiety, and not prone to hydrolysis. Furthermore, the results we present with both of these ligands agree with one another, confirming that the pre-hydrolysis conformation we observe is not just a quirk of one particular analogue.

The referee objects to this assignment, but does not provide any alternative interpretation of the data. Instead, they make spurious arguments against each analogue individually:

They suggest **ATP γ S** might be being hydrolysed over the course of the experiment to the extent that it would lead us to misinterpret our data. This is not plausible in a single molecule setup – even in the case of ATP, let alone a slowly hydrolysing analogue. Our experiment contains 1 μ M SecA, 1 mM ATP, and negligible (30 pM) SecYEG. As the basal turnover rate of SecA is 0.56 min⁻¹ (Robson *et al.* 2009), simple calculations suggest ~34 μ M ATP will be turned over in an hour (60 min * 0.56 min⁻¹ * 1 μ M SecA) – *i.e.* 3.4% of the total ATP. For ATP γ S this figure will be much lower. Consistent with this, there is no change in FRET behaviour over the course of data collection (see attached figure).

The second argument is that ATP γ S is often used: “*to capture [...] ADP, and apo states*”.

This defies reason, and no supporting literature is provided. To capture an ADP-bound state, why add ATP γ S instead of ADP? And to capture the apo state, why add nucleotide at all? We presume that this statement might be based on crystal structures, wherein ATP γ S may not be captured in a bound state (yielding apo), or where the high protein concentrations and long time scales (weeks) of crystallisation may enable hydrolysis to occur. This would definitely NOT be the same for 1 h acquisition in solution under the single molecule conditions we employ.

For **AMPPNP**, which very clearly cannot be hydrolysed, the referee instead claims that the lower affinity of SecA for AMPPNP means it should behave differently to ATP γ S. But we are working at saturating nucleotide concentrations (1 mM, where the affinity is 50 μ M according to Fak *et al.* 2004), so the affinity is irrelevant and therefore the data should, and does, look the same.

The assertion that the data contradict our previous findings seems to be based on direct comparison of two data sets without consideration for the different ways they were obtained and interpreted. In the previous paper (Allen et al., 2016) we were unable to follow fast dynamics and assumed (incorrectly) that changes in SecY occur on the same time scale as those of the SecA ATPase. Based on this assumption and available structural data we interpreted the FRET patterns in terms of three static structural states. However, using better time resolution and the recently developed advanced data processing methodology we show that the data can be explained by just two states with rapid exchange between them. The fact that results of the two studies qualitatively agree is the best that can be expected, given that the mismatch between the number of states and their dynamic nature make direct quantitative comparison impossible.

As none of the above criticisms pass scrutiny, the sensible conclusion is that AMPPNP and ATP γ S do in fact accurately represent the ATP-bound (**pre-hydrolysis**) state – as is widely reported in the literature. From this, it follows that the conformation brought about by addition of ATP (or ADP.AIF_x) can be assigned as a **post-hydrolysis** state (consistent with the standard interpretation of ADP.AIF_x), because it maps neither to the pre-hydrolysis ATP nor ADP bound states.

The second set of comments by referee 3 focus on the PrIA4 variant. They make several assertions, which reflect a miss-understanding of the mechanism of protein transport, which are better addressed point-by-point:

“Concerning the explanation that Topen of PrI4 dictates the translocation process, do the authors think that a single opening event (e.g., within 4 ms) could allow the passage of an entire protein strand?”

As we state in the manuscript, transport is rate limited by passage of bulky and positively charged regions of pre-protein. It is the transport of these regions only (not the entire polypeptide) that require the channel to be open for ~4 ms.

“If not, when a protein peptide is present in the channel without signal sequence peptides (SS), will the FRET signal indicate an open state or a closed state? In other words, do the FRET experiments in this report distinguish between the opening of the lateral gate induced by SS (at the initialization stage, fig.4a), SS and a translocating peptide inside the channel (at the early translocation stage), and a peptide alone in the channel (at the middle and later translocation stages)? The authors do not seem to have considered this difference. However, in the reviewer's opinion, it is crucial to define the relationship between these various "open" states and the FRET signal before proceeding with further analysis.”

Because the signal sequence inserts as a hairpin with the early mature domain of the pre-protein and remains associated with the channel throughout the entire transport process, these are not in fact distinct states.

“Presumably, an idle channel has a closed lateral gate (LG), an active channel with SS and a translocating peptide has an open LG. An active channel with only a translocating peptide has an LG opening somewhere between these two states. Neglecting to address this issue could potentially affect the interpretation of the Topen/Tclosed plots for the ATP+proSpy sample.”

No presumption here is necessary, our data show that this is not the case. The channel is in dynamic equilibrium between two distinct states (open and closed), regardless of nucleotide or pre-protein occupancy. This is the crux of the paper.

We look forward to working with you towards a resolution to the differences between ourselves (and reviewers 1 and 2), to those of reviewer 3.

Your sincerely,

Ian Collinson on behalf of the author team

Channel position over the course of the data collection for ATP γ S, binned into 10 minute intervals. The absence of any time dependence confirms that ATP γ S is not being significantly hydrolysed over the course of the experiment.

Dear Ian,

As you know, of the three referee reports we received on the revised version of your manuscript, two were positive and supported publication. However, referee #3 maintained that there were significant technical barriers and insisted on a substantial re-working of the experimental design in a major revision. The concerns focussed on questions about the suitability of the controls which were used. Your reply contested this, and two sides of a very technical debate became entrenched.

I therefore sought the opinion of a fourth external expert to address the question of whether the scientific conclusions were reliable, considering the nucleotides used. This arbitrating opinion has now concluded that, as the experimental context is a FRET assay that is focused on dynamics, the reviewer's concerns about gamma-S-ATP not strictly describing the pre-hydrolysis state could safely be contextualised. There follows a recommendation to discuss structural studies that indicate that gamma-S-ATP may not be exactly the same state as the pre-hydrolysis state (in a structural sense).

You have, therefore, addressed all concerns satisfactorily and, unless any unexpected issues arise, I will not seek any additional input from the referees.

Before I can finally accept the manuscript though, there are some remaining editorial points which need to be addressed. In this regard would you please:

- acknowledge funding from Leeds Beckett University in our online submission system,
- change the title of the 'conflict of interest' statement to the 'disclosure and competing interests statement',
- upload the figures as separate files, keeping the figure legends in the manuscript file,
- rename the appendix files Appendix Figure S1-S19 and Appendix Table S1-S7 in the table of contents and legends,
- complete the Source Data checklist. Please contact Hannah Sontag if you have any questions about this,
- check links to data referred to in the data availability section are functional,
- indicate the statistical test used for data analysis in the legend of figure 5b,
- in the legend of figure 4c, 'n' and the measure of centre for the error bars need to be defined,
- define error bars in the legends of figures 5b-c,
- although 'n' is provided in the legend to figure 5c, describe the nature of the entity, and
- check the email address for author Daniel W Watkins is valid.

We include a synopsis of the paper (see <http://emboj.embopress.org/>). Please provide me with a two-sentence general summary statement and 3-5 bullet points that capture the key findings of the paper.

We also need a summary figure for the synopsis. The size should be 550 wide by [200-400] high (pixels). You can also use something from the figures if that is easier.

EMBO Press is an editorially independent publishing platform for the development of EMBO scientific publications.

Best wishes,

William

William Teale, PhD
Editor
The EMBO Journal
w.teale@embojournal.org

- a point-by-point response to the referees' comments, with a detailed description of the changes made (as a word file).
- a word file of the manuscript text.
- individual production quality figure files (one file per figure)
- a complete author checklist, which you can download from our author guidelines

(<https://www.embopress.org/page/journal/14602075/authorguide>).

- Expanded View files (replacing Supplementary Information)

We realize that it is difficult to revise to a specific deadline. In the interest of protecting the conceptual advance provided by the work, we recommend a revision within 3 months (1st Feb 2024). Please discuss the revision progress ahead of this time with the editor if you require more time to complete the revisions. Use the link below to submit your revision:

All editorial and formatting issues were resolved by the authors.

Dear Ian,

I am pleased to inform you that your manuscript has been accepted for publication in The EMBO Journal.

Congratulations! I'll be really happy to see this work in our pages.

Best wishes,

William

William Teale, PhD
Editor
The EMBO Journal
w.teale@embojournal.org
